# A Lagrangian convective transport scheme including a simulation of the time air parcels spend in updrafts

Ingo Wohltmann[1], Ralph Lehmann[1], Georg A. Gottwald[2], Karsten Peters[3,*], Alain Protat[4], Valentin Louf[5], Christopher Williams[6], Wuhu Feng[7], and Markus Rex[1]

[1]Alfred Wegener Institute for Polar and Marine Research, Potsdam, Germany
[2]School of Mathematics and Statistics, University of Sydney, New South Wales, Australia
[3]Max Planck Institute for Meteorology, Hamburg, Germany
[4]Bureau of Meteorology, Melbourne, Australia
[5]Monash University, Clayton, Australia
[6]NOAA, Boulder, Colorado, USA
[7]National Centre for Atmospheric Science, School of Earth and Environment, University of Leeds, Leeds LS2 9JT, UK
[*]now at Deutsches Klimarechenzentrum GmbH (DKRZ), Hamburg, Germany

**Correspondence:** I. Wohltmann (ingo.wohltmann@awi.de)

**Abstract.** We present a Lagrangian convective transport scheme developed for global chemistry and transport models, which considers the variable residence time that an air parcel spends in convection. This is particularly important for accurately simulating the tropospheric chemistry of short-lived species, e.g. for determining the time available for heterogeneous chemical processes on the surface of cloud droplets.

In current Lagrangian convective transport schemes air parcels are stochastically redistributed within a fixed time step according to estimated probabilities for convective entrainment as well as the altitude of detrainment. We introduce a new scheme which extends this approach by modelling the variable time that an air parcel spends in convection by estimating vertical updraft velocities. Vertical updraft velocities are obtained by combining convective mass fluxes from meteorological analysis data with a parameterization of convective area fraction profiles. We implement two different parameterizations, a parameterization using an observed constant convective area fraction profile as well as a parameterization which uses randomly drawn profiles to allow for variability. Our scheme is driven by convective mass fluxes and detrainment rates that originate from an external convective parameterization, which can be obtained from meteorological analysis data or from general circulation models.

We study the effect of allowing for a variable time that an air parcel spends in convection by performing simulations, where our scheme is implemented into the trajectory module of the ATLAS chemistry and transport model, and is driven by ECMWF ERA Interim reanalysis data. In particular, we show that the redistribution of air parcels in our scheme conserves the vertical mass distribution and that the scheme is able to reproduce the convective mass fluxes and detrainment rates of ERA Interim. We further show that the estimated vertical updraft velocities of our scheme are able to reproduce wind profiler measurements performed in Darwin, Australia, for velocities larger than $0.6\,\mathrm{m\,s^{-1}}$.

20       $SO_2$ is used as an example to show that there is a significant effect on species mixing ratios when modelling the time spent in convective updrafts compared to a redistribution of air parcels in a fixed time step. Furthermore, we perform long-time global trajectory simulations of radon-222 and compare with aircraft measurements of radon activity.

## 1    Introduction

The parameterization of sub-grid scale cumulus convection and the associated vertical transport is a key procedure in general
circulation models (e.g. Emanuel, 1994; Arakawa, 2004) as well as in chemistry and transport models (e.g. Mahowald et al., 1995). In particular, an accurate simulation of convective transport is important for the modelling of species in chemistry and transport models and would allow for a reduction of uncertainty in the simulation of these species in the troposphere (e.g. Mahowald et al., 1995; Forster et al., 2007; Hoyle et al., 2011; Feng et al., 2011).

      Lagrangian (trajectory-based) models have several advantages over Eulerian (grid-based) models, for example they do not
introduce artificial numerical diffusion and there is no additional computational cost for transporting more than one tracer species (e.g. Wohltmann and Rex, 2009).

      We present a Lagrangian convective transport scheme developed for global chemistry and transport models. The scheme can also be used for applications such as backward trajectories starting along flight paths or sonde ascents, where it allows for simulating the effect of convection when using a statistical ensemble of trajectories starting at every measurement location.
Our convective transport scheme is based on a statistical approach similar to schemes in other Lagrangian models (e.g. Collins et al., 2002; Forster et al., 2007; Rossi et al., 2016). In these schemes air parcels are redistributed vertically within a short fixed time step to simulate the effect of convection. The schemes are driven by convective mass fluxes and detrainment rates derived from a physical parameterization of convection. Typically, the time period between entrainment and detrainment is assumed to be fixed in these schemes, and varies between 15 minutes and 30 minutes in Collins et al. (2002), Forster et al. (2007) and
Rossi et al. (2016). The fixed convective time step is not necessarily the same as the advection time step.

      These schemes therefore do not take into account the variable residence times of air parcels inside a convective cloud. The amount of time spent inside the cloud is particularly important when considering the tropospheric chemistry of short-lived species. The concentrations of these species in the upper troposphere may crucially depend on the transport time of an air parcel from the boundary layer to the upper troposphere (e.g. Hoyle et al., 2011). An example for a species for which this
is relevant is the short-lived species $SO_2$, which is depleted by a range of fast heterogenous reactions inside clouds and by a gas-phase reaction with OH (e.g. Berglen et al., 2004; Tsai et al., 2010; Rollins et al., 2017).

      Therefore, we extend the approach of earlier schemes by simulating the variable residence time air parcels spend inside a convective cloud by estimating vertical updraft velocities. Vertical updraft velocities are obtained from combining convective mass fluxes from meteorological analysis data with a parameterization of convective area fraction profiles. The scheme is
implemented into the trajectory module of the ATLAS chemistry and transport model (e.g. Wohltmann and Rex, 2009) and simulations are performed which are driven by ECMWF ERA Interim reanalysis data (Dee et al., 2011).

We test the scheme for the conservation of the vertical mass distribution and for reproducing the convective mass fluxes and detrainment rates of the meteorological analysis used for driving the model. Particular emphasis is given to the study of different methods of parameterizing the convective area fraction profiles needed to simulate vertical updraft velocities. All of these tests are performed with idealized trajectory simulations which ignore the large-scale wind fields to facilitate interpretation.

In addition, global long-time trajectory simulations which use the large-scale wind fields are performed. These include simulations of radon-222 which are compared to aircraft measurements and the simulation of an artificial tracer that is designed to imitate the most important characteristics of $SO_2$ chemistry.

Radon-222 is widely used to validate convection models and to evaluate tracer transport (e.g. Feichter and Crutzen, 1990; Mahowald et al., 1995; Jacob et al., 1997; Forster et al., 2007; Feng et al., 2011). Radon is removed entirely by radioactive decay, and hence, no uncertainties in chemistry, microphysics or deposition have to be considered. Furthermore, the half-life time of 3.8 days is in the right order of magnitude to detect changes by transport on short time scales. However, meaningful conclusions from the validation runs are limited due to uncertainties in radon emissions and the relatively sparse coverage of radon measurements. In addition, the globally constant lifetime of radon prevents a validation of the parameterization of the time spent in convective updrafts, which would only be possible with a varying lifetime.

When considering convective transport of a $SO_2$-like tracer in a global simulation we see a significant impact of the variable residence time on mixing ratio profiles, compared to a scheme with a redistribution of air parcels in a fixed time step.

The outline of the paper is as follows: Section 2 and Section 3 describe the convective transport scheme and the corresponding algorithm. Section 2 describes the modelling of entrainment, upward transport, detrainment, and subsidence outside of clouds. Section 3 describes the method to calculate vertical updraft velocities. In Section 4, the performance of our scheme is tested. The conservation of the vertical mass distribution and the reproduction of the mass fluxes and detrainment rates from meteorological analysis data are examined, global trajectory-based simulations of radon-222 are compared to measurements, and simulated vertical updraft velocities are compared with wind profiler measurements from Darwin, Australia. In Section 5, simulations of a $SO_2$-like tracer are shown to demonstrate that using the scheme can have a significant effect on tracer mixing ratios. We conclude with a discussion and summary in Section 6.

## 2 Description of the convective transport scheme

### 2.1 General concept

We first present the algorithm for forward trajectories, and introduce the necessary adaptions for backward trajectories at the end to facilitate understanding.

A statistical approach is taken, where entrainment and detrainment probabilities are calculated for each trajectory at every time step. Whether a given trajectory air parcel is entrained into a cloud or detrained from a cloud is then determined by drawing random numbers. The model is driven by convective mass fluxes and detrainment rates provided by meteorological analysis data or by general circulation models. Typical resolutions of meteorological analysis data are of the order of $1° \times 1°$. A

grid box of the analysis typically contains several convective systems which only affect a small fraction of the mass contained
in the grid box, which necessitates a statistical approach.

We extend the approach used in existing convective transport schemes by allowing for a variable time that an air parcel spends inside the convective event. To determine this time, vertical updraft velocities are calculated by combining convective mass fluxes from meteorological analysis data with parameterizations of convective area fraction profiles (a detailed account is given in Section 3). Instead of calculating the probability that an entrained air parcel detrains at a certain altitude and then
redistributing the parcels accordingly in a fixed time step (as in the approach of Collins et al., 2002, or Forster et al., 2007), an advection time step of the trajectory model is divided into smaller intermediate convective time steps of a few seconds, and the parcel is moved upwards and tested for detrainment in each intermediate convective time step.

Our algorithm executes the following steps for each trajectory air parcel in every advection time step $\Delta t$ of the trajectory model (typically 10 minutes):

1. Entrainment if air parcel is not in convection and if a test for entrainment is successful (Section 2.3)

2. If the air parcel takes part in convection, the following two steps are repeated with a smaller intermediate convective time step $\Delta t_{\mathrm{conv}}$ of 10 seconds until the air parcel detrains or the end of the present advection time step of the trajectory model $\Delta t$ is reached:

   – Upward transport by the distance given by the convective time step $\Delta t_{\mathrm{conv}}$ multiplied by the vertical updraft
velocity (Section 2.4)

   – Detrainment if a test for detrainment is successful (Section 2.5)

3. Subsidence of air parcels outside of convection in the environment (Section 2.6)

The advection time step of the trajectory model $\Delta t$ needs to be sufficiently short for the algorithm to work (see Sections 2.3 and 2.5).

The Lagrangian convective transport model is driven by convective mass fluxes and detrainment rates from meteorological analysis data or from general circulation models and thus relies on an external convective parameterization. The convective mass flux $M(z)$ at a given location, geometric altitude $z$ and time in units of mass transported per area and per time interval is related to the entrainment rate $E(z)$ and the detrainment rate $D(z)$ by mass conservation

$$\frac{\mathrm{d}M}{\mathrm{d}z} = E - D \tag{1}$$

where $E$ and $D$ are given in units of mass per area, per time interval and per vertical distance. Both $E$ and $D$ are defined as positive numbers.

In meteorological analysis data, the atmosphere is divided into several model layers. Usually, the convective mass flux is given at the layer interfaces, while the detrainment rates are given as the mean values of the layers. Entrainment rates can be calculated from the mass fluxes and detrainment rates using Equation 1. In addition, the atmosphere is divided into grid boxes
with a given horizontal resolution. In the ERA Interim meteorological reanalysis, $M$ is given as the grid-box mean convective

updraft mass flux and $D$ as the grid-box mean updraft detrainment rate per geometric altitude. The convective mass flux $M$ is related to the mean convective mass flux in the convective updrafts $M_{up}$ (per area of updraft) by

$$M = f_{up}M_{up} \tag{2}$$

where $f_{up}$ is the convective area fraction, which is the fraction of the area of the grid box covered by updrafts in convective clouds. We will only consider updrafts here, since updraft mass fluxes typically dominate over downdraft mass fluxes in the clouds (see e.g. Figure 3 in Kumar et al., 2015, or Collins et al., 2002). It is planned to simulate downdraft mass fluxes in a future version of the model.

## 2.2 The mass of trajectory air parcels

In the following, it is assumed that every trajectory air parcel is associated with a mass, which is equal to the mass of the other trajectory air parcels and is constant in time. While there is no natural way to assign a mass to a single trajectory air parcel, this is different in a global model, where the model domain is filled with trajectory air parcels. One could argue that an air parcel only refers to an infinitesimally small volume and that only intensive quantities such as density are well defined for a trajectory air parcel, while extensive quantities such as mass are not well defined. However, in a global model, the volume of the model domain can be divided into smaller subvolumes that make up the complete volume. Each subvolume can be associated with a trajectory air parcel, and the air parcel mass is given as the product of the density of air and the air parcel volume. The same constant mass can be assigned to each trajectory air parcel, which implies that the associated volume is increasing with decreasing air density. Since the subvolumes of air parcels should not overlap to avoid that the same air volume is counted twice, this implies that the trajectory air parcels need to be distributed uniformly over pressure (but exponentially decreasing over altitude).

This is not merely a theoretical consideration, but becomes important when e.g. the total mass of a chemical species is calculated, or the mass flux of a chemical species through a control surface (as the tropopause).

The mass of a trajectory air parcel in such a model is typically much larger than the mass transported in a single convective event (e.g. Collins et al., 2002). For this reason and due to the statistical nature of the approach, results are only meaningful if a sufficiently large ensemble of trajectories is examined before interpreting the results. The equations of the scheme are independent of the mass associated with the trajectory. Thus, in a global model where trajectory air parcels fill the model domain, a larger mass associated with a particular trajectory air parcel (corresponding to a lower density of parcels per volume) leads to a lower number of trajectory air parcels in convection at a given point in time, which balances the higher mass moved per convective event.

## 2.3 Entrainment

To model the entrainment of the trajectory air parcels we follow the approach of Collins et al. (2002) and Forster et al. (2007) and assume that the atmosphere is divided into several layers, where layer $k$ is confined by levels $k$ and $k+1$, as illustrated in Figure 1. These layers may be identified with the model layers of the meteorological analysis. For an air parcel located in a

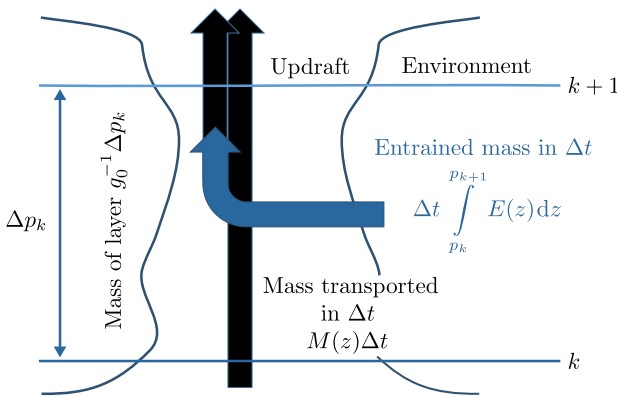

**Figure 1.** Schematic representation of the entrainment step. All quantities are per unit area.

layer between pressures $p_k$ and $p_{k+1}$, the probability $\varepsilon$ of it being entrained in an advection time step of the trajectory model $\Delta t$ is defined by the ratio of the mass per area entrained in a layer in a time step $\Delta t$ and the mass per area of the layer. The entrainment probability is independent of the area covered by convection and is given by

$$\varepsilon = \frac{g_0 \Delta t \int_{z(p_k)}^{z(p_{k+1})} E \, \mathrm{d}z}{\Delta p_k} \quad \text{with} \quad \Delta p_k = p_k - p_{k+1} \tag{3}$$

where $g_0$ is the gravitational acceleration of the Earth and $\int E \, \mathrm{d}z$ is the grid-box mean entrainment rate integrated over the layer (resulting in the same units as the convective mass flux). The integration has to be performed over geometric altitude, which requires a conversion between pressure and geometric altitude.

Whether an air parcel is entrained and takes part in convection is decided by generating a uniformly distributed random number $r_{\mathrm{entr}}$ in the interval $[0, 1]$ in every trajectory time step and comparing that to the calculated probability. If the random number is smaller than the entrainment probability $r_{\mathrm{entr}} < \varepsilon$, the air parcel is marked as taking part in convection and is therefore not tested for being entrained as long as it stays in convection. The advection time step of the trajectory model $\Delta t$ needs to be sufficiently short to avoid that $\varepsilon > 1$ (which would mean that the air in the layer would be ventilated several times by convection during the advection time step $\Delta t$).

The time of the entrainment event can be anywhere in the time interval between $t$ and $t + \Delta t$. For simplicity, we assume that the convective event always starts at time $t$. This only results in a small shift of the convective event by a few minutes at most (depending on the advection time step), which will be negligible in most cases.

## 2.4 Upward transport

If a parcel is marked as taking part in convection, it is transported upwards for the vertical distance that it will be able to ascend in one intermediate convective time step $\Delta t_{\mathrm{conv}}$ (10 seconds). The vertical distance is determined by the vertical convective updraft velocity. After the intermediate convective time step, the parcel is tested for detrainment (see Section 2.5). This procedure is repeated until either the test for detrainment is successful or the end of the present advection time step of the

trajectory model $t + \Delta t$ is reached. The short intermediate convective time step $\Delta t_{\mathrm{conv}}$ is necessary to capture the steep vertical gradients in the detrainment rates and convective mass fluxes. For a strong updraft of $10\,\mathrm{m\,s^{-1}}$, a time step of $10\,\mathrm{s}$ corresponds to a vertical distance of $100\,\mathrm{m}$, which is usually sufficient to resolve the vertical levels of the analyses.

The vertical updraft velocity inside the convective cloud is determined by noting that the convective mass flux in the cloud is the product of density and the vertical updraft velocity $M_{\mathrm{up}} = \rho w_{\mathrm{up}}$, where the density is given by $\rho = p/(RT)$ according to the ideal gas law, where $R = 287\,\mathrm{J\,kg^{-1}\,K^{-1}}$ is the specific gas constant of dry air (neglecting modifications of $R$ due to water vapour) and $T$ is temperature. Using Equation 2 the vertical updraft velocity inside the convective cloud (in units of geometric altitude per time) is given by

$$w_{\mathrm{up}} = \frac{MRT}{f_{\mathrm{up}}p} \tag{4}$$

All quantities are interpolated to the position of the air parcel.

Neither convective area fractions $f_{\mathrm{up}}$ nor vertical updraft velocities $w_{\mathrm{up}}$ are usually available from meteorological analysis data. To overcome this problem in our convection scheme we estimate profiles of the convective area fraction $f_{\mathrm{up}}$ based on observations. We implement two methods here: The first method uses an observed constant climatological convective area fraction profile, while the second uses a stochastic parameterization for randomly drawn convective area fraction profiles (Gottwald et al., 2016). A detailed discussion of the calculation of the vertical updraft velocities is given in Section 3.

Once the vertical updraft velocity $w_{\mathrm{up}}$ is determined, the vertical geometric distance $\Delta z_{\mathrm{conv}}$ that the air parcel ascends in an intermediate convective time step $\Delta t_{\mathrm{conv}}$ is given by

$$\Delta z_{\mathrm{conv}} = w_{\mathrm{up}}\Delta t_{\mathrm{conv}} \tag{5}$$

Under the assumption that the coordinate system of the trajectory model is log-pressure height $Z$, the distance that the parcel ascends in log-pressure height is

$$\Delta Z_{\mathrm{conv}} = \Delta z_{\mathrm{conv}}\frac{T_0}{T} \tag{6}$$

where log-pressure height is defined as $Z = -H\log(p/p_0)$ and $H = RT_0/g_0$. $T_0$ and $p_0$ are the reference temperature and reference pressure of the log-pressure coordinate. Other coordinate systems will require equivalent transformations. The new vertical location of a trajectory air parcel is determined by adding $\Delta Z_{\mathrm{conv}}$ to the initial vertical position of the parcel. The longitude and latitude of the parcel remain unchanged.

## 2.5 Detrainment

If a parcel is marked as taking part in convection and has been transported upwards, it is tested next for detrainment.

The probability that a parcel is detrained during an intermediate convective time step $\Delta t_{\mathrm{conv}}$ can be determined by noting that air involved in convection in the layer defined by $\Delta z_{\mathrm{conv}}$ (regardless whether it had been entrained in that layer or whether it had been transported from below) can only leave via two paths: either it can be detrained or it can leave through the upper boundary. Thus, the detrainment probability is the ratio of the amount of air that is detrained between the start and end position

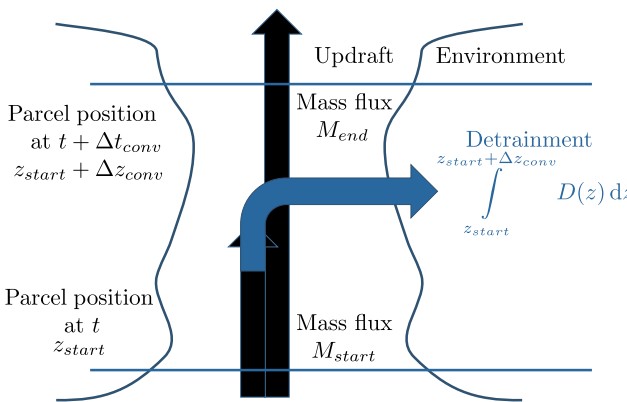

**Figure 2.** Schematic representation of the detrainment step. All quantities are per unit area.

of the air parcel and the sum of the amount of air entering either from below or through entrainment between the start and end position. Assuming that air coming from below behaves the same way as entrained air and that there is no preferred pathway out of the layer for air coming from below or for entrained air, the detrainment probability is given by

$$\delta = \frac{\int_{z_{\text{start}}}^{z_{\text{start}}+\Delta z_{\text{conv}}} D\,\mathrm{d}z}{M_{\text{start}} + \int_{z_{\text{start}}}^{z_{\text{start}}+\Delta z_{\text{conv}}} E\,\mathrm{d}z} \tag{7}$$

or, equivalently

$$\delta = \frac{\int_{z_{\text{start}}}^{z_{\text{start}}+\Delta z_{\text{conv}}} D\,\mathrm{d}z}{M_{\text{end}} + \int_{z_{\text{start}}}^{z_{\text{start}}+\Delta z_{\text{conv}}} D\,\mathrm{d}z} \tag{8}$$

where $M_{\text{start}}$ is the convective mass flux at the start position of the air parcel and $z_{\text{start}}$ is the altitude of the start position. $z_{\text{start}} + \Delta z_{\text{conv}}$ is the end position of the air parcel after one intermediate convective time step (see Figure 2). Conversions from the coordinate system of the trajectory model to geometric altitude are necessary here.

Whether the air parcel is detrained and leaves convection is decided by generating a uniformly distributed random number $r_{\text{detr}}$ and comparing that to the calculated probability $\delta$. If the random number is smaller than the detrainment probability $r_{\text{detr}} < \delta$, the parcel leaves the convection at altitude

$$z_{\text{detr}} = z_{\text{start}} + \Delta z_{\text{conv}} \frac{r_{\text{detr}}}{\delta} \tag{9}$$

Multiplication with $r_{\text{detr}}/\delta$ ensures that the detrainment heights are uniformly distributed in $[z_{\text{start}}, z_{\text{start}}+\Delta z_{\text{conv}}]$. Assuming that the air parcel always leaves at $z_{\text{start}} + \Delta z_{\text{conv}}$ would overestimate the detrainment altitude systematically, since $\delta$ is the probability that the parcel detrains somewhere between $z_{\text{start}}$ and $z_{\text{start}}+\Delta z_{\text{conv}}$. A parcel is allowed to entrain and detrain in the same advection time step $\Delta t$ (but can stay longer in convection, of course).

The approach for detrainment described above differs from the approach employed in previous Lagrangian convective transport schemes, since it takes into account the explicit simulation of the time that air parcels spend in convective updrafts, whereas

schemes such as those employed in Collins et al. (2002) or Forster et al. (2007) assume a constant time that parcels spend in convection. The probability that an entrained air parcel detrains at a given altitude, however, is the same in both approaches.

If the parcel reaches an altitude where the convective mass flux $M$ interpolated to the position of the parcel is zero, but still has not detrained, the parcel is forced to detrain. Due to the finite time step, the air parcel may end up at a position where $M = 0$, which can be interpreted as numerical overshooting. While this behaviour can be avoided by decreasing the altitude of the parcel until $M > 0$, we do not correct for this, since the correction is typically less than $100\,\mathrm{m}$.

If the air parcel detrains before reaching the end of the present advection time step $\Delta t$ of the trajectory model, it cannot entrain again until the start of the next advection time step. A correction can be applied to account for the time missing for new entrainment between the detrainment event (which is at some intermediate convective time step $\Delta t_{\mathrm{conv}}$) and the start of the next advection time step. This can be accomplished by adding the missing time to the $\Delta t$ of the next entrainment test of the trajectory. The effect of this correction is usually small, provided the advection time step is sufficiently small.

The size of the advection time step $\Delta t$ is crucial. Since the trajectory model generates outputs only every $\Delta t$ time units, the trajectory is marked as detrained only after the next advection time step and not after at the intermediate time step. If the advection time step is too large, chemical reactions may be overestimated inside of convective clouds.

## 2.6 Subsidence outside of convective systems

To conserve mass and balance the updraft, parcels in the environmental air have to subside. All parcels that are currently not in convection are moved downwards by a pressure difference of

$$\Delta p_{\mathrm{subs}} = \frac{1}{1 - f_{\mathrm{up}}} g_0 M \Delta t \tag{10}$$

where $M$ and $f_{\mathrm{up}}$ are the convective mass flux and convective area fraction, respectively, interpolated to the position of the trajectory air parcel. The factor $1/(1 - f_{\mathrm{up}})$ accounts for trajectory air parcels which are in convection rather than subsiding. Note that this factor is close to 1 since $f_{\mathrm{up}} \approx 10^{-3}$. The fraction of trajectory air parcels which are taking part in convection does not necessarily correlate with $f_{\mathrm{up}}$, which is based on observations independent from the convective parameterization driving the model. However, the results of the validation runs show that the conservation of the vertical mass distribution of the runs is not noticeably affected by this uncertainty (see Section 4).

Alternatively, the fraction of trajectory air parcels that are currently in convection in the model run could be used. This is however only possible for global runs. The mass flux of trajectories through a given surface is not necessarily balanced for non-global ensembles of trajectories. The approach would require to average the results over a volume that is small enough to allow for variations in the fraction, but large enough to contain a sufficient number of air parcels.

Another alternative would be to subside all air parcels and not only the air parcels, which are currently not in convection (Collins et al., 2002). Subsiding air parcels which are currently in convection is however not only unphysical, but also can result in air parcels that descend while they are in convection, and possibly detrain at a lower altitude than they were entrained.

## 2.7 Backward trajectories

An attractive feature of the algorithm is that it can be readily employed for backward trajectories. Backward trajectories with convection are useful for e.g. determining the source regions of air measured along a flight path or sonde ascent and modelling their chemical composition.

The following modifications of the algorithm are necessary. First, the meaning of $E$ and $D$ in the equations has to be exchanged (detrainment becomes entrainment backwards in time). Moreover, the "updraft" velocity $w_{up}$ has to be applied with a negative sign. Finally, the correction for subsidence moves the air parcels upward. The "entrainment" probabilities from Equation 3 are now "detrainment probabilities backwards in time", and are given by

$$\varepsilon = \frac{g_0 \Delta t \int_{z(p_k)}^{z(p_{k+1})} D \, dz}{\Delta p_k} \quad \text{with} \quad \Delta p_k = p_k - p_{k+1} \tag{11}$$

Analogously, the "detrainment" probabilities become "entrainment probabilities backwards in time" with

$$\delta = \frac{\int_{z_{start}-\Delta z_{conv}}^{z_{start}} E \, dz}{M_{start} + \int_{z_{start}-\Delta z_{conv}}^{z_{start}} D \, dz} \tag{12}$$

In contrast to forward trajectories, the convective mass flux at the start position of the air parcel $M_{start}$ is at a higher altitude $z_{start}$ than the end position $z_{start} - \Delta z_{conv}$.

If the parcel reaches either an altitude where $M = 0$ or propagates below the surface (due to the finite time step), but still has not "detrained" the parcel is forced to "detrain".

## 3 Determining vertical updraft velocities

Vertical updraft velocities can be calculated by using Equation 4. Except for the convective area fraction $f_{up}$, all quantities can be obtained from meteorological analysis data. We implement two methods to estimate $f_{up}$, which are described in Sections 3.1 and 3.2.

### 3.1 Constant convective area fraction

The first method uses a constant climatological profile $f_{up}(z)$ of the convective area fraction, which is derived from observations. The variability of the vertical velocities is dominated by the variability of the convective mass flux $M$ for a constant convective area fraction profile (see Equation 4).

The constant convective area profile used in the method is shown in Figure 3. The profile resembles the profile in Figure 2 of Kumar et al. (2015) (red lines using the "space approach", estimating the fraction of convection by comparing the area of convective precipitation to the total measured area). This profile was obtained using C-band dual polarization (CPOL) precipitation radar measurements conducted in Darwin, Australia during two wet seasons (2005/2006 and 2006/2007), and is representative for a 190 x 190 $km^2$ grid box centered over Darwin.

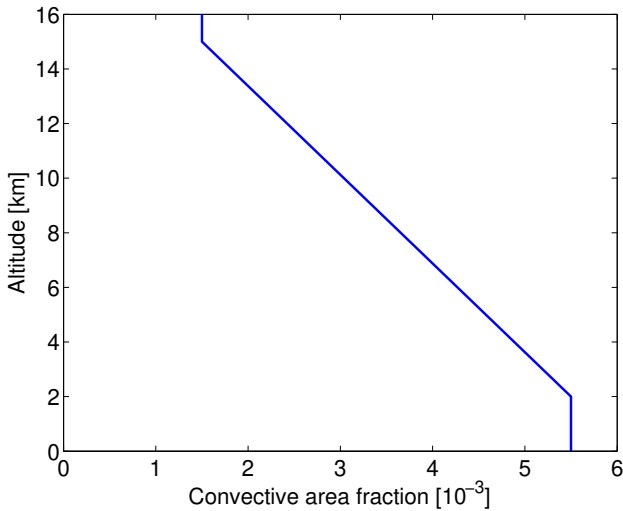

**Figure 3.** Constant convective area fraction profile used for calculating vertical updraft velocities.

The scanning area of the radar is comparable to typical grid sizes of meteorological analysis data. Kumar et al. (2015) show that the measured mean convective area fraction is independent of the observed area for a wide range of values (from a circle of radius 10 km to a circle of radius 100 km).

Our scheme was originally developed for an application in the tropics. Note that an application of the algorithm in the extratropics would require a different convective area fraction profile. We present simulations for the tropics as well as global long-time simulations of radon-222 in Sections 4 and 5. The global simulations however, are not sensitive to the choice of the convective area fraction profile due to the globally constant lifetime of radon (see Section 4.4.4). Hence, using a tropical profile in the radon runs does not noticeably change the results compared to a run using a profile for the mid-latitudes.

To account for variable convective area fraction profiles as observed in measurements, we now implement a second method.

## 3.2 Random convective area fraction

The second method uses a stochastic parameterization of the convective area fraction to obtain randomly drawn convective area fraction profiles and was introduced by Gottwald et al. (2016). The method is based on estimates of convective area fractions derived from CPOL radar measurements over Darwin (wet seasons 2004/2005, 2005/2006, 2006/2007, Davies et al., 2013) and Kwajalein, Marshall Islands (May 2008 to January 2009), averaged over 6 hours. The parameterization depends on the large-scale vertical velocity at 500 hPa as an input parameter. The large-scale vertical velocity at 500 hPa was derived by Davies et al. (2013) by variational analysis using ECMWF operational analysis data constrained by area-mean surface precipitation from the CPOL instrument. Frequency distributions of the convective area fraction are derived from the CPOL measurements as a function of the large-scale vertical velocity at 500 hPa. Figures 1a and 1b of Gottwald et al. (2016) show the resulting frequency distribution for Darwin and Kwajalein, respectively.

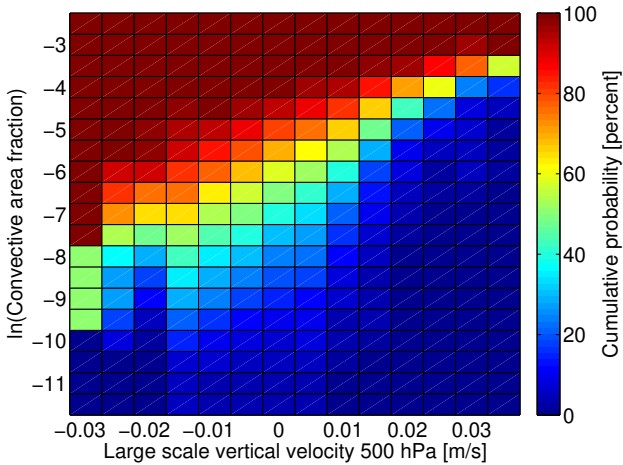

**Figure 4.** Cumulative frequency distribution of the natural logarithm of the convective area fraction from a combined Darwin/Kwajalein CPOL radar dataset as a function of the large-scale vertical velocity at 500 hPa. The distribution is used to calculate the vertical updraft velocities in the algorithm.

We combine the Darwin and Kwajalein data into one data set to increase the number of measurements. Peters et al. (2013) and Gottwald et al. (2016) have shown that the functional dependency of convection on the large-scale vertical velocity at 500 hPa is sufficiently similar at both locations.

To derive the frequency distribution used in this study, the combined data are binned into a 2-dimensional lookup table, which uses bins for the large-scale vertical velocity and bins for the natural logarithm of convective area fraction. The logarithm is used to obtain a more uniform distribution over the bins. The resulting lookup table is shown in Figure 4. The data are binned in $0.005 \, \mathrm{m\,s^{-1}}$ ($1.2 \, \mathrm{hPa\,h^{-1}}$) bins ranging from $-0.035 \, \mathrm{m\,s^{-1}}$ to $0.04 \, \mathrm{m\,s^{-1}}$ for the large-scale vertical velocity and in $0.5$ bins ranging from $-12$ to $-2$ for the natural logarithm of the convective area fraction. For values of the large-scale vertical
velocity greater than $0.04 \, \mathrm{m\,s^{-1}}$ (smaller than $-10.2 \, \mathrm{hPa\,h^{-1}}$), we use the deterministic relationship $f_{\mathrm{up}} = 0.8807v$ obtained by linear regression ($v$ large-scale vertical velocity in $\mathrm{m\,s^{-1}}$), as done in Gottwald et al. (2016).

The large-scale vertical velocity of ERA Interim at 500 hPa interpolated to the position of the trajectory air parcel is used to select one of the vertical velocity bins of the frequency distribution. A uniformly distributed random number is drawn to determine a value for the convective area fraction from the lookup table. This value is used as the convective area fraction at
cloud base. To obtain a vertical profile, the value is then scaled with a normalized version of the profile from Kumar et al. (2015) described in Section 3.1. The scaling with a constant profile ensures that the resulting profile of vertical updraft velocities will be physically reasonable (in contrast to a method where the vertical updraft velocity would be obtained independently at every level). The vertical updraft velocities are then determined from the convective area fractions using Equation 4.

Due to the stochastic character of the method, it is unavoidable that unrealistic vertical updraft velocities are produced from
time to time. To prevent unrealistically large values, vertical velocities larger than $20 \, \mathrm{m\,s^{-1}}$ are reset to $20 \, \mathrm{m\,s^{-1}}$. Similarly,

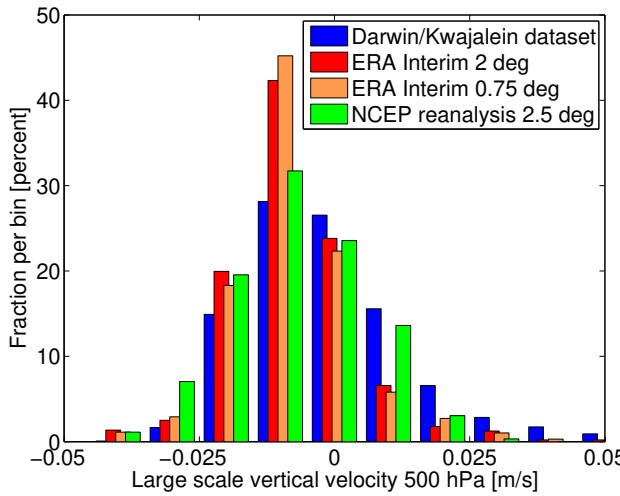

**Figure 5.** Frequency distribution of the vertical velocities at $500\,\mathrm{hPa}$ of the Darwin/Kwajalein dataset compared to frequency distributions of the vertical velocity from the ERA Interim reanalysis ($0.75°$ x $0.75°$ and $2°$ x $2°$ horizontal resolution) and the NCEP reanalysis ($2.5°$ x $2.5°$ horizontal resolution). For the reanalysis data, the vertical velocity at $500\,\mathrm{hPa}$ at all grid points between $180°$ E and $240°$ E and $30°$ S and $30°$ N (Pacific Ocean) for the arbitrary date 1 June 2010, $00\,\mathrm{h}$ UTC is used. Bin width is $0.01\,\mathrm{m\,s^{-1}}$.

values smaller than $0.1\,\mathrm{m\,s^{-1}}$ are reset to $0.1\,\mathrm{m\,s^{-1}}$ to avoid that the trajectory air parcels remain in convection for too long. We checked that this procedure only affects at most a few percent of the trajectories.

### 3.2.1  Dependency of the stochastic parameterization on the large-scale wind fields and the horizontal resolution

We tacitly assume here that the large-scale vertical velocities of the Darwin/Kwajalein dataset, which are used to determine
the convective area fraction profile, and those of the reanalysis are comparable. It is known that differences exist for the large-scale vertical velocities of different reanalysis datasets, which in addition depend on the horizontal resolution of the reanalysis (e.g. Monge-Sanz et al., 2007; Hoffmann et al., 2019). Figure 5 shows the frequency distribution of the vertical velocities at $500\,\mathrm{hPa}$ of the Darwin/Kwajalein dataset compared to frequency distributions of the vertical velocity from the ERA Interim reanalysis ($0.75°$ x $0.75°$ and $2°$ x $2°$ horizontal resolution) and the NCEP reanalysis ($2.5°$ x $2.5°$ horizontal resolution, Kistler
et al., 2001). For the reanalysis data, the distribution of the large-scale vertical velocity at $500\,\mathrm{hPa}$ at all grid points between $180°$ E and $240°$ E and $30°$ S and $30°$ N is shown (Pacific Ocean). The frequency distributions of all four datasets (including the different horizontal resolutions) agree sufficiently well and differences are acceptable in view of other uncertainties of our method, e.g. the uncertainties of the convective area fraction. Hence, we did not apply a scaling or other correction to the large-scale vertical velocities from ERA Interim. To apply our method to different reanalysis datasets, their vertical velocities
at $500\,\mathrm{hPa}$ would need to be compared to those of the Darwin/Kwajalein data set, and potentially have to be shifted or scaled to obtain a realistic distribution of the convective area fractions.

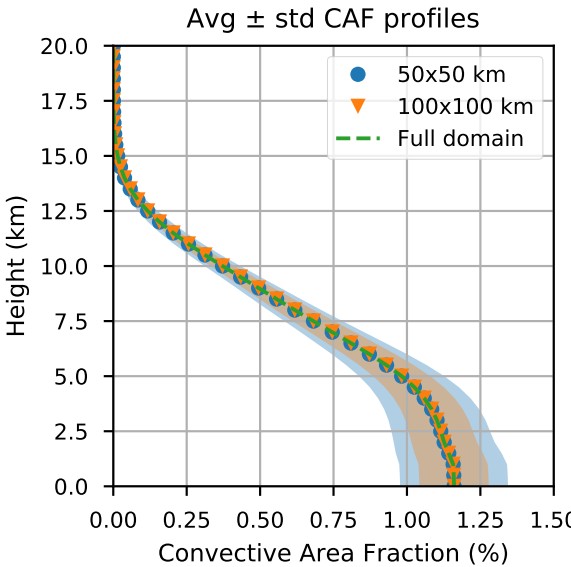

**Figure 6.** Dependence of the standard deviation of the frequency distribution of measured convective area fractions on the used domain size of the CPOL radar. Shaded areas show the standard deviation for a domain size of 190 x 190 $\text{km}^2$ (green), 100 x 100 $\text{km}^2$ (red) and 50 x 50 $\text{km}^2$ (blue). For the smaller domain sizes, the measurement domain of the radar has been divided into smaller subdomains. The shaded areas give the standard deviation. The green line shows the mean convective area fraction.

The frequency distribution of the measured convective area fractions depends on the size of the measured area from which the frequency distribution is derived. We use the full domain size of the radar of 190 x 190 $\text{km}^2$, which is comparable to a horizontal resolution of the meteorological analysis of about 2° x 2°. The domain size should be comparable to the grid size of the meteorological analysis data to obtain a meaningful distribution of vertical updraft velocities. Smaller domain sizes may produce significant differences in the distribution. As the domain size decreases, the frequency distribution tends to approximate a bimodal distribution: grid cells completely covered by convection and grid cells completely free of convection become more frequent (e.g. Arakawa and Wu, 2013).

Figure 6 shows the dependence of the standard deviation of the frequency distribution of measured convective area fractions on the used domain size of the CPOL radar. Results are shown for domain sizes of 190 x 190 $\text{km}^2$, 100 x 100 $\text{km}^2$ and 50 x 50 $\text{km}^2$. For the smaller domain sizes, the measurement domain of the radar is divided into smaller subdomains. The shaded areas give the standard deviation. It is evident that the frequency distributions for different domain sizes differ significantly. The current implementation of the algorithm does not consider this effect, and it is not clear if incorporating a distribution of the convective area fractions which depends on the grid size would lead to a significant change of the results of the trajectory runs. An implementation of frequency distributions of the convective area fractions that depend on grid size is planned for a future version.

### 3.3 Limitations and possible alternatives

A limitation of our stochastic parameterization to derive $f_{\mathrm{up}}$ is that we do not take into account the convective mass flux at the position of the trajectory air parcel. Ideally, we would like to use the convective mass flux as the large-scale variable for the stochastic parameterization of the convective area fractions and as a replacement for the large-scale vertical velocity at $500\,\mathrm{hPa}$. This, however, requires observations of convective mass fluxes, which can only be obtained from simultaneous measurements of convective area fractions and updraft velocities (see Kumar et al., 2015).

Alternatively to our approach to estimate the vertical updraft velocity via the convective area fraction and using Equation 4, one might use a climatological profile of measured mean vertical updraft velocities. However, this has the disadvantage that the shape of the wind profile is always the same. To obtain variability in the vertical updraft velocities, a random scaling could be applied to the wind profile. Measurements of updraft velocities are available from in situ aircraft observations (e.g. LeMone and Zipser, 1980), airborne Doppler radar (e.g. Heymsfield et al., 2010) or ground-based wind profilers (e.g. May and Rajopadhyaya, 1999; Kumar et al., 2015). We tested this method with a mean vertical velocity profile taken from Schumacher et al. (2015), but found that the convective area fractions implied from the vertical velocity profile and the convective mass fluxes of the meteorological analysis (cf. Equation 4) were greater than 1 in some altitudes. This issue is equivalent to the issue of the unrealistic vertical updraft velocities in the methods described above using the convective area fractions. A correction for the unrealistic convective area fractions in the approach using a climatological profile of vertical updraft velocities turned out to be more difficult than a correction for the unrealistic vertical updraft velocities in the approach using observations of convective area fractions.

## 4 Performance of the convective transport scheme

We examine the performance of our Lagrangian convective transport model by testing the conservation of the vertical mass distribution and the reproduction of the convective mass fluxes and detrainment rates of the meteorological analysis in an idealized trajectory simulation, which ignores the large-scale wind fields. Within the same idealized setup, we show that our method yields vertical updraft velocities which are consistent with observations of velocities larger than $0.6\,\mathrm{m\,s^{-1}}$. We further show results on the residence time of trajectory air parcels in convection. Long-time global trajectory simulations of radon-222, which use the large-scale wind fields, are compared to measurements and global simulations of an artificially designed short-lived $SO_2$-like tracer are used to explore how allowing for variable residence times affects the model results.

For all of these simulations, we perform trajectory runs driven by meteorological data of the ECMWF ERA Interim reanalysis (Dee et al., 2011) with $0.75° \times 0.75°$ or $2° \times 2°$ horizontal resolution, which include large-scale wind fields, temperature, updraft convective mass fluxes, detrainment rates and boundary layer heights. Large-scale winds and temperatures are used with $6\,\mathrm{h}$ temporal resolution, while convective mass fluxes, detrainment rates and boundary layer heights are used with $3\,\mathrm{h}$ resolution to capture the diurnal cycle. Entrainment rates are not provided by ECMWF and are calculated from the detrainment rates and convective mass fluxes using Equation 1. The convective parameterization of the ERA Interim reanalysis in the underlying IFS model is originally based on the scheme of Tiedtke (1989), with several modifications (e.g. Bechtold et al.,

2004). The trajectory module is the same that is used in the ATLAS chemistry and transport model (Wohltmann and Rex, 2009), extended for the convective transport scheme. A 4th order Runge-Kutta scheme is used for calculating the trajectories. For this study, only the trajectory module of the ATLAS model is used, the detailed chemistry scheme and mixing scheme of the model are not needed in the model runs (see Section 4.4.1).

  While the quality of the convective mass fluxes and detrainment rates will have a large impact on the results of the radon
validation and the validation of the vertical updraft velocities, it is out of the scope of this study to give a validation of ERA Interim. We refer the reader to the existing literature here (e.g. Dee et al., 2011; Taszarek et al., 2018).

## 4.1 Conservation of vertical mass distribution and reproduction of convective mass fluxes and detrainment rates

For an initial technical verification of the algorithm, we test the conservation of the vertical mass distribution and examine if our scheme appropriately reproduces the convective mass fluxes and detrainment rates of the reanalysis. We use an idealized
setup here to facilitate the interpretation.

  In the idealized setup, we start 100,000 trajectories that are initially uniformly distributed in pressure between $1000\,\mathrm{hPa}$ and $100\,\mathrm{hPa}$ and are uniformly distributed horizontally between $180°\,\mathrm{E}$ and $240°\,\mathrm{E}$ and $30°\,\mathrm{S}$ and $30°\,\mathrm{N}$ (Pacific Ocean). We impose a horizontal domain without topography to simplify interpretation. The Pacific Ocean is chosen since we are mainly interested in applying our model for tropical convection. Each trajectory is assigned a constant mass corresponding
to the volume it occupies. The runs are driven by temporally constant convective mass fluxes and detrainment rates from ERA Interim ($0.75°$ x $0.75°$ horizontal resolution) taken from the arbitrarily chosen date 1 June 2010, $00\,\mathrm{h}$ UTC. Large-scale horizontal and vertical winds are set to zero. That is, trajectory air parcels can only move vertically by convection inside the cloud or subsidence outside of the cloud. Trajectory air parcels which propagate below the surface due to the finite time step are lifted above the surface. The trajectory model uses a log-pressure coordinate. Trajectories are run for 20 days with an advection
time step $\Delta t$ of 10 minutes. Four different runs are performed for forward and backward trajectories combined with the two vertical updraft velocity parameterizations described in Sections 3.1 and 3.2. Figure 7 shows arbitrarily selected trajectories from the forward run when the constant convective area fraction profile is used.

  Figure 8 shows the conservation of the vertical mass distribution for forward trajectories when the constant convective area fraction profile described in Section 3.1 is used. The number of the trajectories in $50\,\mathrm{hPa}$ bins at the end of the run (red)
compares well to the number of trajectories in these bins at the start of the run (blue). There is only a small deviation at the lowest levels caused by the fact that all trajectories are initialized with pressures smaller than $1000\,\mathrm{hPa}$, whereas ERA Interim also features larger values of the surface pressure. This causes some trajectories to end at pressures larger than $1000\,\mathrm{hPa}$. Results for backward trajectories and results employing the random convective area fraction profile described in Section 3.2 look very similar.

In the idealized setup, a significant fraction of the trajectory air parcels does not move at all, because they are initialized at a position where the convective mass flux and entrainment rate are zero. The number of these trajectories is shown in black in Figure 8. A more rigorous test of conservation of the vertical mass distribution with a long-time simulation driven by the actual large-scale wind fields is presented in Section 4.4.3.

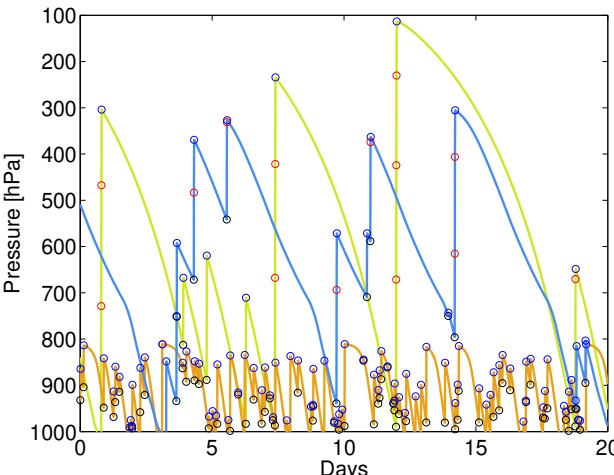

**Figure 7.** Example trajectories from the run with the idealized setup for forward trajectories with large-scale wind set to zero and constant convective area profile. Open black circles mark entrainment, open red circles upward transport in convection in 10 minute steps and open blue circles detrainment.

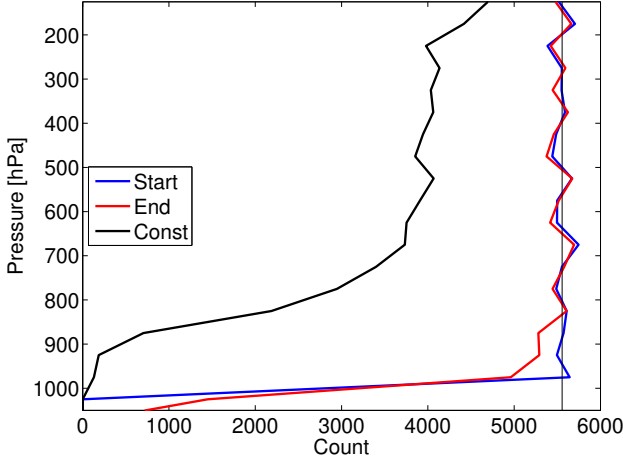

**Figure 8.** Conservation of vertical mass distribution after 20 days for forward trajectories and using a constant convective area fraction profile. Number of trajectories in $50\,\mathrm{hPa}$ bins at the start of the run (blue) and at the end of the run (red). The black line denotes the number of trajectories that did not move due to zero convective mass flux at their start position.

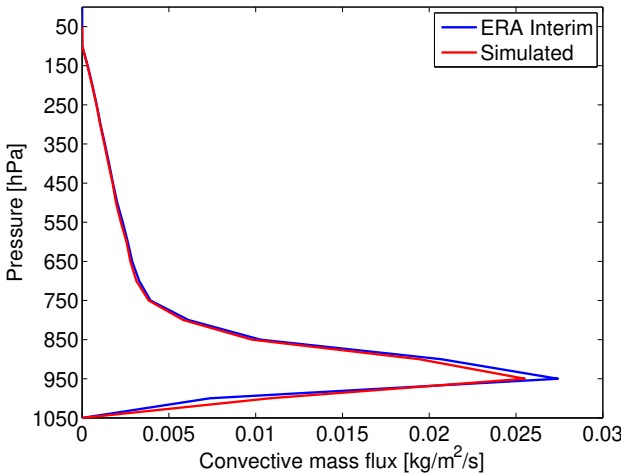

**Figure 9.** Mean convective mass flux profile from ERA Interim compared to the simulated convective mass flux profile for forward trajectories and using a constant convective area fraction profile (in a region from $180°$ E to $240°$ E and $30°$ S to $30°$ N, 20 days with meteorological fields of 1 June 2010, 00 h UTC).

Figure 9 shows the mean convective mass flux profile from ERA Interim averaged over the tropical domain described above

compared with the simulated mass flux profile for forward trajectories using the constant convective area fraction profile. Simulated mass fluxes are calculated by counting the trajectory air parcels that pass a given pressure level during one advection time step and which are in convection at this time. The number of the trajectories is multiplied by the air parcel mass and divided by the area of the tropical domain and the time period of 20 days. The agreement between ERA Interim and the simulations is very good. There is only a slight underestimation of the pronounced maximum around 950 hPa. Again, results

for backward trajectories and results employing the random convective area fraction profile described in Section 3.2 look very similar.

Figure 10 shows the same for the detrainment rates. Detrainment rates are calculated by counting the trajectory air parcels which experience detrainment in a given pressure layer during one advection time step. The number of these detrained trajectory air parcels is multiplied by the air parcel mass, divided by the area of the tropical domain, the time period of 20 days and the

mean vertical extent in geometrical altitude of the pressure layer. Again, agreement is very good and results for backward trajectories or for random convective area fraction profiles look very similar.

While the mean convective mass flux and the detrainment rate profiles are insensitive to the choice of the convective area fraction profile, we see in the following section that the vertical updraft velocity profiles strongly depend on whether a constant convective area profile or a randomly drawn profile is implemented.

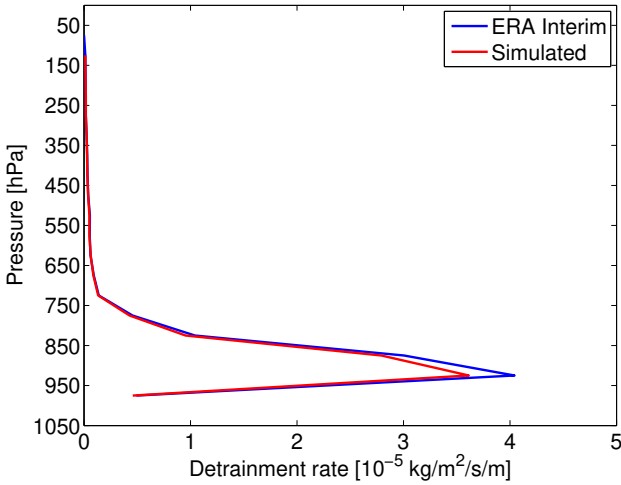

**Figure 10.** Mean detrainment rate profile from ERA Interim compared to the simulated detrainment rate profile for forward trajectories and constant convective area fraction profile (in a region from $180°$ E to $240°$ E and $30°$ S to $30°$ N, 20 days with meteorological fields of 1 June 2010, 00 h UTC).

### 4.2 Validation of the vertical updraft velocities with wind profiler measurements

We validate the modelled vertical updraft velocities against wind profiler measurements. The modelled vertical updraft velocities are taken from the idealized forward trajectory runs in the tropical Pacific from Section 4.1. Results for backward trajectories are very similar.

The modelled velocities are compared with measurements from a 50- and 920-MHz wind profiler pair situated in Darwin, Australia. The time resolution of the measurements is 1 minute and vertical updraft velocities are obtained by the method of Williams (2012). Data comprise the wet seasons 2003/2004, 2005/2006, 2006/2007 and 2009/2010. Cloud top heights are determined from the $0\,\mathrm{dBz}$ echo top height of the CPOL radar instrument at Darwin. The field of view of this instrument covers the wind profiler site. Convective profiles are identified by using only wind profiler measurements, where the CPOL instrument shows convective precipitation. CPOL data are available every 10 minutes. All wind profiler measurements within $\pm 5$ minutes of the CPOL measurement times are considered and cut at the corresponding cloud top height.

Figure 11 shows frequency distributions of the vertical updraft velocities binned in $0.2\,\mathrm{m\,s^{-1}}$ bins for selected $50\,\mathrm{hPa}$ pressure bins. The frequency distributions of the vertical updraft velocities from the Darwin measurements are shown in black, modelled distributions employing the constant convective area fraction profile are shown in magenta and modelled distributions employing random convective area fraction profiles are shown in red. The solid lines show the distributions when vertical updraft velocities smaller than $0.6\,\mathrm{m\,s^{-1}}$ are excluded, while the dashed lines show distributions comprised of all velocity values.

There is a large number of measurements with small vertical updraft velocities. The sensitivity of the measured distributions to these small values is quite large, and the measured distributions excluding values smaller than $0.6\,\mathrm{m\,s^{-1}}$ differ significantly from the measured distributions which incorporate all values. The distributions obtained from our scheme show considerably less values smaller than $0.6\,\mathrm{m\,s^{-1}}$ and there is less of a difference between the modelled distributions when all velocities or only those larger than $0.6\,\mathrm{m\,s^{-1}}$ are accounted for.

It is difficult to assess the reasons for the marked disagreement between model and measurements in the small vertical updraft velocities. The number of small values is sensitive to the method to determine convective situations in the wind profiler measurements, and may change significantly depending on the method. It is common to apply a lower threshold to the vertical updraft velocities to define convective situations (e.g. LeMone and Zipser, 1980; May and Rajopadhyaya, 1999; Kumar et al., 2015). Typically, this threshold is between $0\,\mathrm{m\,s^{-1}}$ and $1.5\,\mathrm{m\,s^{-1}}$ and may have a significant effect (see discussion in Kumar et al., 2015). Hence, part of the disagreement can be attributed to the conceptual problem of defining what a convective updraft is.

For the modelled profiles, the distribution of the velocities is determined by a large number of factors and may change significantly depending on the details of implementation and the convective parameterization in the underlying meteorological analysis. For example, the assumed convective area fraction profile and the assumptions in the Tiedtke scheme plays a large role. Hence, we do not expect more than a qualitative agreement between model and measurements, in particular for small updraft velocities. The lower threshold of $0.1\,\mathrm{m\,s^{-1}}$ implemented into our convective transport scheme (see Section 3.2) should however play no role in Fig. 11, since the bin width is $0.2\,\mathrm{m\,s^{-1}}$.

The distribution of the vertical updraft velocities reproduces the distribution of the measurements fairly well, when only velocities greater than $0.6\,\mathrm{m\,s^{-1}}$ are considered. In particular, the magnitude of the approximately exponential decrease in the frequency distribution is met well.

In the case when random convective area fraction profiles are employed our method yields a higher frequency of large vertical velocities compared to the case when the constant convective area fraction profile is implemented. The random convective area fraction profile method leads to a better agreement with observations. In particular, the two implementations differ significantly for values of the vertical updraft velocity larger than $5\,\mathrm{m\,s^{-1}}$.

The fact that the vertical updraft velocities are typically larger when a randomly drawn convective area fraction profile is used can be readily understood qualitatively: Assuming that $M$, $T$ and $p$ are fixed, the mean updraft velocity in case of a mean constant convective area fraction profile $\langle f_{\mathrm{up}} \rangle$ is simply $\langle w_{\mathrm{up1}} \rangle = \frac{MRT}{\langle f_{\mathrm{up}} \rangle p}$, where $\langle \ldots \rangle$ denotes the mean over all air parcels. In the case of a varying randomly drawn convective area fraction profile, the mean vertical updraft velocities need to be expressed as $\langle w_{\mathrm{up2}} \rangle = \langle \frac{MRT}{f_{\mathrm{up}} p} \rangle = \frac{MRT}{p} \langle \frac{1}{f_{\mathrm{up}}} \rangle$. Since $\langle \frac{1}{f_{\mathrm{up}}} \rangle \geq \frac{1}{\langle f_{\mathrm{up}} \rangle}$ due to the fact that the harmonic mean is always smaller than the geometric mean, we obtain the relation $\langle w_{\mathrm{up2}} \rangle \geq \langle w_{\mathrm{up1}} \rangle$ between the mean vertical updraft velocities of the two implementations. This implies that also individual realizations of $w_{\mathrm{up}}$ are on average larger for the random convective area fraction profiles.

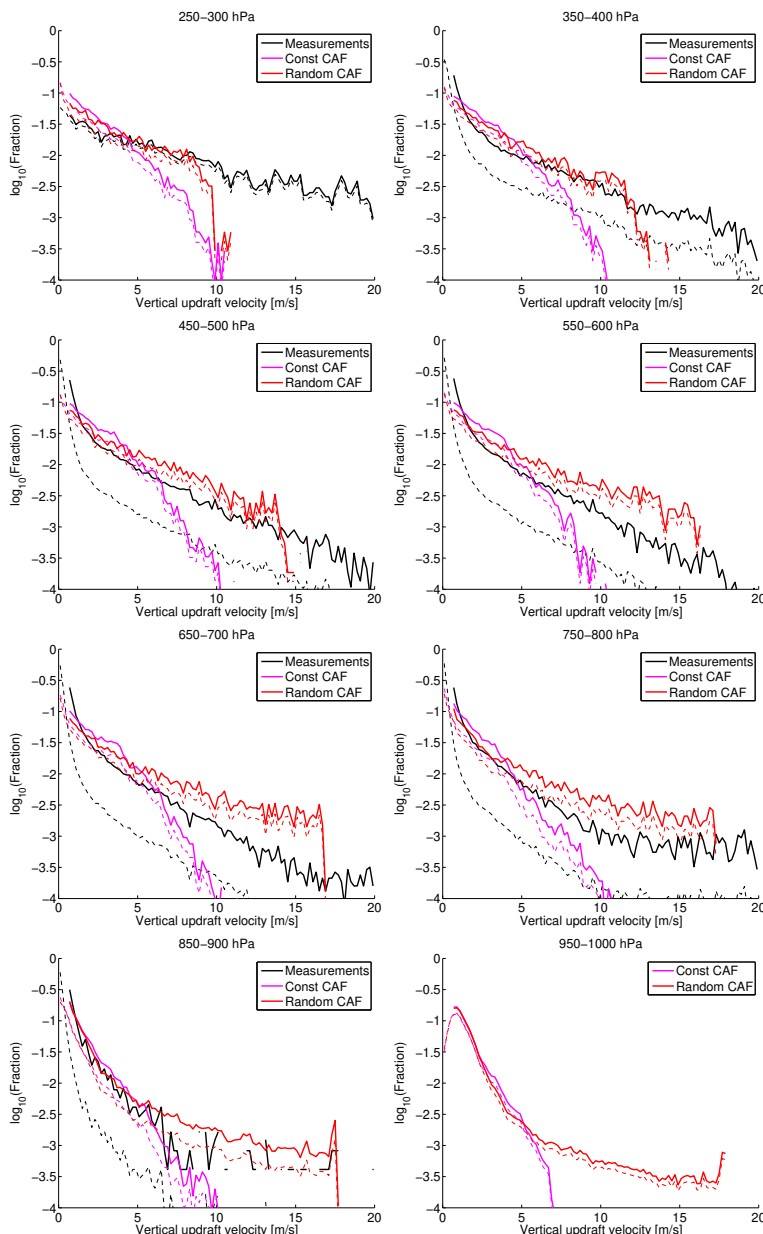

**Figure 11.** Frequency distribution of vertical updraft velocities for different pressure bins from wind profiler measurements in Darwin, Australia, in $0.2\,\mathrm{m\,s^{-1}}$ bins (black), compared to the corresponding frequency distributions of vertical updraft velocities obtained from the constant and random convective area fraction profile method (magenta and red). The dashed lines show the distribution including all velocity values ($> 0\,\mathrm{m\,s^{-1}}$), for the solid lines all values below $0.6\,\mathrm{m\,s^{-1}}$ are excluded.

Replacing the simulated vertical updraft velocities by the measured vertical updraft velocities in the model (including values smaller than $0.6\,\mathrm{m\,s^{-1}}$) would increase the average residence time between entrainment and detrainment. In turn, this would lead to a lower concentration of a short-lived species like $SO_2$ in the upper troposphere.

The model is trained on convective area fraction data measured in Darwin and Kwajalein and compared to wind profiler data measured at Darwin, while it is applied to a larger region covering a large part of the tropical Pacific here. The lack of other measurements does not allow for a completely independent model validation.

## 4.3 Residence time in convection

Figure 12 shows the frequency distribution of the residence times of the trajectories between entrainment and detrainment obtained from simulations employing both parameterizations for the vertical updraft velocity (solid lines). Most convective events have a residence time of less than 30 minutes (more than 95 % when the constant convective area fraction profile is implemented). Since the number of convective events is dominated by shallow convective events, which typically only lift the air parcel a few hundred meters in one advection time step (cf. Figure 7), we also show the frequency distribution for deep convection (dashed lines), defined here by detrainment events above $300\,\mathrm{hPa}$. These will be more relevant when considering the upper tropospheric mixing ratio of short-lived species. Typical residence times of deep convective events are estimated to be about 1 hour when the constant convective area fraction profile is implemented. The simulation using random convective area fraction profiles yields a higher number of convective events with a short residence time and correspondingly, a lower number of convective events with long residence times, compared to the simulation using the constant convective area fraction profile. This is consistent with the larger simulated vertical updraft velocities when using randomly generated convective area fraction profiles.

## 4.4 Comparison of long-time simulations of radon-222 with aircraft measurements

Long-time global trajectory simulations of radon-222 are compared here with aircraft observations. The results depend to a great extent on the used meteorological data. They are presented here to demonstrate that the model is able to produce reasonable results with a given meteorological analysis.

Radon-222 is formed by the radioactive decay of uranium in rock and soils and has been widely used to validate convection models and to evaluate tracer transport (e.g. Feichter and Crutzen, 1990; Mahowald et al., 1995; Jacob et al., 1997; Collins et al., 2002; Forster et al., 2007; Feng et al., 2011). It is chemically inert, is not subject to wet and dry deposition and is only removed by radioactive decay. Hence, its removal processes are very well known. The half-life time of 3.8 days is in the right order of magnitude to detect changes in convective transport. However, the measurement coverage of radon is quite limited (in particular for profiles) and emissions are uncertain (e.g. Liu et al., 1984; Mahowald et al., 1995). Furthermore, the globally constant lifetime of radon does not allow for any validation of the parameterization of the vertical updraft velocities. Nevertheless, radon-222 is currently widely used for validation of convective transport due to a lack of alternatives.

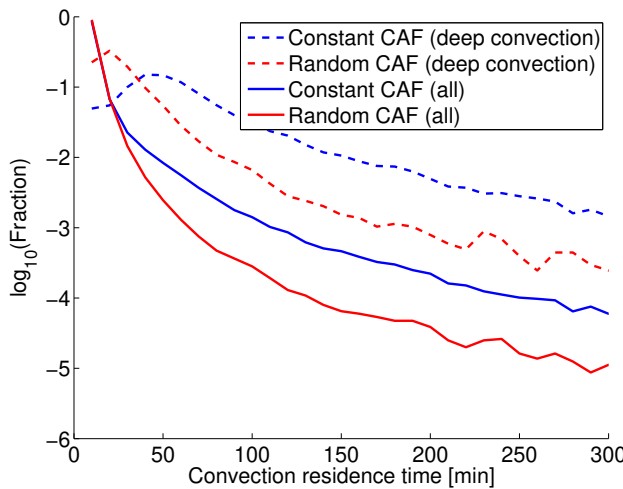

**Figure 12.** Frequency distribution of the residence times of the trajectories between entrainment and detrainment simulated by the two parameterizations for the vertical updraft velocity. The fraction of all events with a given duration is shown in 10 minute bins. Solid lines show the distribution for all convective events, while dashed lines show the contribution from deep convective events (detrainment above 300 hPa).

### 4.4.1 Setup of the radon runs

Global runs are performed for the time period 1 January 1989 to 31 December 2005. Trajectories are initialized at random positions (both horizontally and in pressure) between 1100 hPa and 50 hPa. The number of trajectories is chosen in such a way that the mean horizontal distance of the trajectories is 150 km in reference to a layer of a width of 50 hPa. The random positioning is the default initialization in ATLAS and avoids that an initialization on a regular grid can have any systematic effects on the results. Trajectories initialized below the surface are discarded. The trajectory model uses a log-pressure coordinate and is driven by ERA Interim data with a horizontal resolution of $2° \times 2°$. The advection time step $\Delta t$ is set to 30 minutes. The change from 10 minutes to 30 minutes and from $0.75° \times 0.75°$ to $2° \times 2°$ is due to computational constraints. We performed 1-year test runs with a $0.75° \times 0.75°$ resolution, a 10 minute time step and a mean horizontal distance of 75 km of the trajectories that show that the results of the run with the lower horizontal and time resolution are nearly identical.

Trajectory air parcels which propagate below the surface due to the finite time step are lifted above the surface. In the uppermost layer (100 hPa to 50 hPa), the positions of the trajectory air parcels are reinitialized to random positions at every time step. There is no special treatment of the boundary layer except for the assumption of a well-mixed layer when distributing the radon emissions. We do not apply any mixing of air parcels to simulate diffusion, contrary to the stratospheric version of the model (Wohltmann and Rex, 2009). Given the resolution of the model runs and the short half-life time of radon, we believe that these simplifications are justified.

Note that the convective area fraction profile used (see Fig. 3) is only appropriate for the tropics. However, the radon runs are not sensitive to the convective area fraction profile due to the globally constant lifetime of radon (see the discussion in Section 4.4.4).

### 4.4.2   Radon emissions

We use the same radon emissions as e.g. Jacob et al. (1997) and Feng et al. (2011). Radon is emitted almost exclusively over land. The radon emissions are $1.0\,\mathrm{atoms\,cm^{-2}\,s^{-1}}$ over land between $60°$ S–$60°$ N, $0.005\,\mathrm{atoms\,cm^{-2}\,s^{-1}}$ over oceans between $60°$ S–$60°$ N, $0.005\,\mathrm{atoms\,cm^{-2}\,s^{-1}}$ between $60°$ and $70°$ in both hemispheres. There is no emission between $70°$ and the poles. These emissions are considered to be accurate on a global scale to within $25\,\%$ and on a regional scale to about a factor of 2 (Jacob et al., 1997; Forster et al., 2007). Radon is emitted into all trajectory air parcels that are in the boundary layer by assuming a well-mixed boundary layer, and a volume mixing ratio $x$ of

$$x = \frac{e\,\Delta t}{\Delta z_{\mathrm{BL}}}\frac{k_B T}{p} \tag{13}$$

is added to each air parcel in the boundary layer, where $e$ is the emission in atoms per area and time interval, $\Delta t$ is the advection time step of the trajectory model, $k_B = 1.38 \cdot 10^{-23}\,\mathrm{J\,K^{-1}}$ is the Boltzmann constant and $\Delta z_{\mathrm{BL}}$ is the local height of the boundary layer. The boundary layer height is provided by ERA Interim.

To avoid large horizontal areas in which no trajectory air parcels receive radon emissions, a minimum boundary layer height of $500\,\mathrm{m}$ is used. The factor $1/\Delta z_{\mathrm{BL}}$ would still ensure mass conservation if no minimum boundary layer height is assumed: the decreasing number of air parcels that receive emissions in a given area when decreasing the height of the boundary layer is balanced by the increasing concentration in the fewer parcels that receive emissions. However, the uptake of emissions by trajectories would become patchy and the horizontal resolution of the emission fields would not be fully used. This is especially relevant for species with strongly spatially varying emissions like $SO_2$.

Our approach may cause some radon which would be trapped in the boundary layer to be emitted immediately into the free troposphere and may cause some differences of the simulation to the radon measurements. However, assuming a minimum boundary layer height (or some similar measure) is unavoidable, since the required number of trajectories needed for a model run which resolves the boundary layer by far exceeds currently available computational capabilities.

### 4.4.3   Conservation of vertical mass distribution

We revisit the issue of the conservation of the vertical mass distribution in this more realistic setup, compared to the idealized setup in Section 4.1. Figure 13 shows the conservation of the vertical mass distribution of air (not of radon) of the long-time simulation. The number of trajectory air parcels in $50\,\mathrm{hPa}$ bins at the end of a run with convection and the constant convective area fraction profile (magenta) compares very well with the number of trajectory air parcels at the start of the run (cyan) and the results of a run without convection (red and blue). The lower number of trajectory air parcels in the bins near the surface is due to orography. The trajectory air parcels remain homogeneously distributed in the horizontal domain without clustering or forming gaps over the course of the model run, confirming that no further measures are required to redistribute trajectories.

#### 4.4.4 Comparison with measurements

We compare the simulations to the climatological mid-latitude profiles of Liu et al. (1984), which have been widely used to validate tracer transport in global models in the past (e.g. Feichter and Crutzen, 1990; Jacob et al., 1997; Collins et al., 2002; Feng et al., 2011). These observations were obtained from aircraft measurements at different continental locations in the northern midlatitudes from 1952 to 1972. Figure 14 shows the simulated mean radon profile for June to August over land (30° N–60° N) compared to the Liu et al. (1984) mean measurement profile for the same season (from 23 sites, bars show

standard deviation of the profiles). Simulation results are averaged over all 15 years of the long-time run, but the years are not identical to the years of measurement, since there is no meteorological data from ERA Interim for this time period. Figure 15 shows the same for December to February (7 sites, no standard deviation available).

Furthermore, we show a comparison of our simulated radon activity to aircraft campaign measurements from coastal locations around Moffett Field (37.5° N, 122° E, California) in June and August 1994 (Kritz et al., 1998) in Figure 16 and a

comparison with aircraft measurements from coastal regions in Eastern Canada (Nova Scotia) from the North Atlantic Regional Experiment (NARE) campaign in August 1993 (Zaucker et al., 1996) in Figure 17. Simulation results are averaged over the campaign periods and over a longitude-latitude bounding box encompassing all aircraft measurements.

The runs with convection generally show higher radon concentrations than the runs without convection in the middle and upper troposphere due to the fast transport of radon from the boundary layer to the detrainment level. A more detailed inter-

pretation of the profiles is however difficult due to the large-scale horizontal averaging.

The agreement of the simulations to the measurements is reasonable, given the large uncertainties in measurements and emissions. While the runs with convection agree better with the measurements than the runs without convection, there are still significant differences. For the same radon measurements, differences of a similar order of magnitude are also observed in other studies and for other convective transport models (e.g. Collins et al., 2002; Forster et al., 2007; Feng et al., 2011).

There is an underestimation of radon by the simulations in the middle troposphere, which is most pronounced in the Moffett Field data (Figure 16), consistent with previous studies (e.g. Jacob et al., 1997; Forster et al., 2007). This may be due to uncertainties in emission and due to the fact that measurements from coastal areas are included, where horizontal radon gradients are high and difficult to model (see also Forster et al., 2007).

The results for both vertical updraft velocity parameterizations are nearly identical because of the globally constant lifetime

of radon. A globally constant lifetime implies that for an air parcel in a given layer, only the time since the last contact with the boundary layer matters and not the exact path that the trajectory air parcel has taken to the layer: It makes no difference if a trajectory air parcel was transported slowly upwards from the emission in the boundary layer to 10 km in the last 10 days or if it was first transported quickly by convection to 10 km within one hour, and then stayed at 10 km for 9 days and 23 hours. For the same reason, a convective redistribution of air parcels with a fixed time step as in Collins et al. (2002) leads to similar

results. Hence, it is not possible to give a recommendation for one of the vertical updraft velocity parameterizations from the results of the radon simulations.

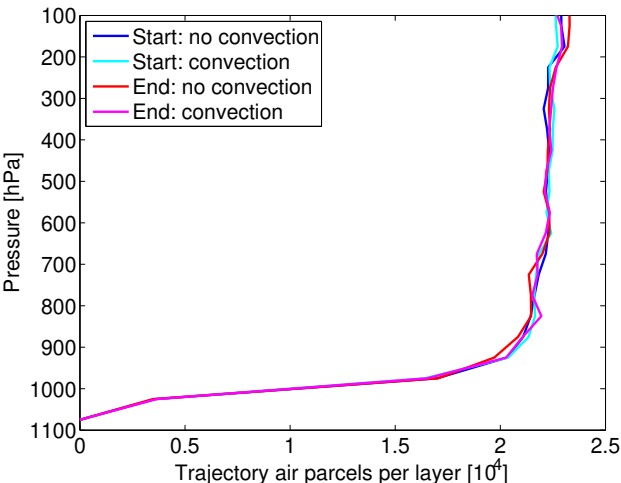

**Figure 13.** Long-time conservation of the vertical mass distribution after 15 years for a run with forward trajectories using the constant convective area fraction profile and for a run without convection. We show the number of trajectory air parcels in 50 hPa bins at the start of the run (blue and cyan) and at the end of the run (red and magenta).

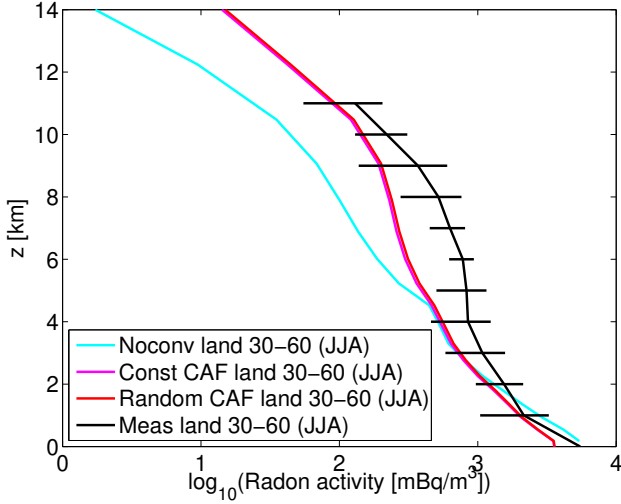

**Figure 14.** Observed mean radon profile obtained from measurements over land (30° N–60° N, June–August) by Liu et al. (1984) compared to the simulated radon obtained from 15 year long-time runs for the same region and months. Bars show the standard deviation of the profiles.

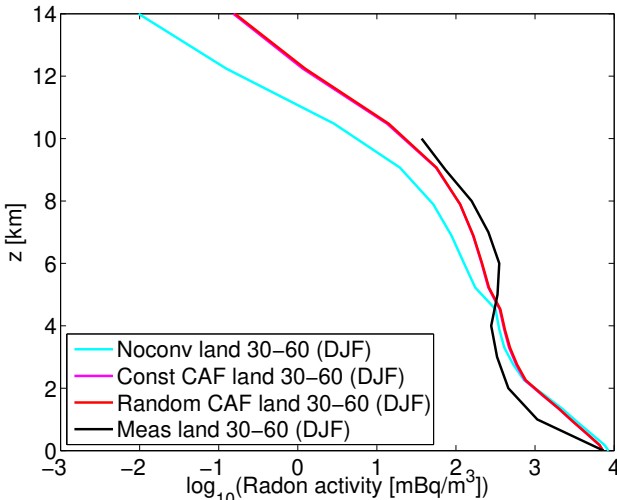

**Figure 15.** Observed mean radon profile obtained from measurements over land (30° N–60° N, December–February) by Liu et al. (1984) compared to the simulated radon obtained from 15 year long-time runs for the same region and months.

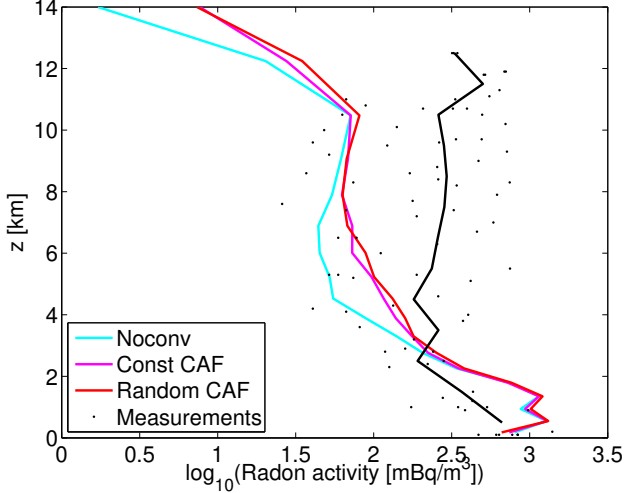

**Figure 16.** Observed radon from aircraft measurements of the Moffett Field campaign (California) in June 1994 (Kritz et al., 1998) compared to the simulated radon from our model in the same time period using a bounding box including all measurements. Dots show the single measurements and the solid black line the mean in 1 km bins.

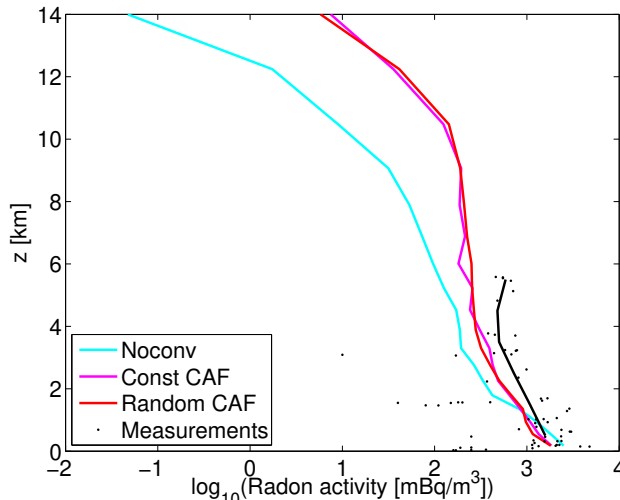

**Figure 17.** Observed radon from aircraft measurements of the North Atlantic Regional Experiment (NARE) campaign in August 1993 (Zaucker et al., 1996) compared to the simulated radon from our model in the same time period using a bounding box including all measurements. Dots show the single measurements and the solid black line the mean in 1 km bins.

## 5 Simulations with a $SO_2$-like tracer

We demonstrate that there is a benefit to explicitly simulate the vertical updraft velocity and to account for a variable time spent in convective clouds, by performing runs with an artificial tracer that is designed to imitate the most important characteristics of the short-lived species $SO_2$, which unlike radon has a varying lifetime (a detailed model of $SO_2$ chemistry and emissions is complex and is outside the scope of this study). $SO_2$ transported from the troposphere to the stratosphere is one of the most important contributors to the stratospheric aerosol layer in volcanically quiescent periods (see e.g. the review in Kremser et al., 2016). In addition, $SO_2$ is a pollutant mainly produced by anthropogenic sources, which is responsible for atmospheric acidification and for the direct and indirect aerosol effect (e.g. Feichter et al., 1996; Berglen et al., 2004; Tsai et al., 2010).

$SO_2$ is depleted by a gas-phase reaction with OH and by several fast heterogenous reactions in the liquid phase in clouds, mainly with $H_2O_2$ (see e.g. Berglen et al., 2004; Tsai et al., 2010; Rollins et al., 2017). The lifetime with respect to the OH reaction is of the order of days to weeks (e.g. Rex et al., 2014), while the lifetime in the presence of clouds is of the order of hours to days (e.g. Lelieveld, 1993). Hence, we perform runs with an artificially designed tracer which has a lifetime of 0.1 days when in convection and of 10 days when not in convection. Emissions are distributed uniformly over the globe. The advection time step of the trajectory model is 10 minutes. The horizontal resolution of ERA Interim is $0.75° \times 0.75°$ and only one year is simulated.

Four different runs are performed: a run without convection, a run with a constant convective area fraction profile, a run with random convective area fraction profiles and a run where the vertical updraft velocity is set to a constant value of $100 \, \mathrm{m \, s^{-1}}$ (with $\Delta t_{\mathrm{conv}}$ set to $1 \, \mathrm{s}$) to mimic the redistribution of trajectory air parcels in a short fixed time step as in previous Lagrangian

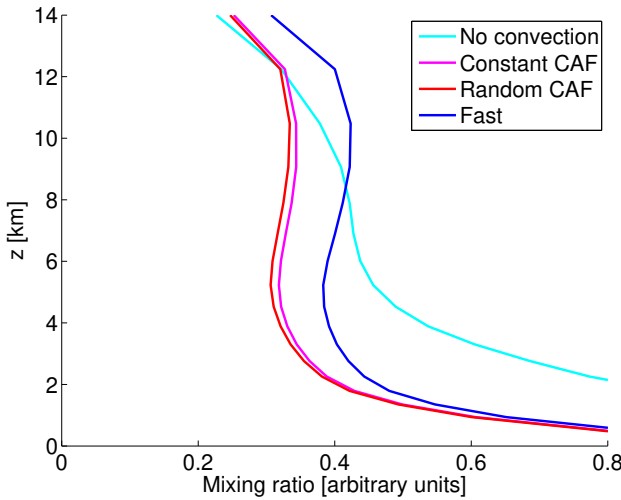

**Figure 18.** Mean simulated artificial $SO_2$-like tracer profiles in the tropics (30° S–30° N) for a run without convection, a run with a constant convective area fraction profile, a run with a random convective area fraction profile and a run where the vertical updraft velocity is set to a constant value of $100\,\mathrm{m\,s^{-1}}$ to mimic the redistribution within a short fixed time step in other Lagrangian convective transport schemes.

convective transport schemes (e.g. Collins et al., 2002). For chemical species with a varying lifetime such as $SO_2$, different vertical updraft velocity parameterizations lead to significantly different tracer concentrations. Such short-lived species are often difficult to validate with measurements. This is due to the large uncertainties in the chemistry schemes and microphysics for these species, uncertain emissions and sparse measurement coverage (see discussion in e.g. Forster et al., 2007).

Figure 18 shows the mean simulated $SO_2$-like tracer profiles in the tropics (30° S–30° N) for the four different runs. The
run without convection leads to larger values of the mixing ratio than the other runs in the lower troposphere, since without convection the tracer is depleted with a long lifetime of 10 days, whereas with convection fast depletion occurs in the convective clouds, leading to a smaller mixing ratio. Conversely, in the upper troposphere, the run without convection yields lower values of the mixing ratio than the runs involving convection, since without convection it takes much longer for a trajectory air parcel to be transported to the upper atmosphere. Residence times in the clouds are shortest in the run where we set the vertical updraft
velocity to the large value of $100\,\mathrm{m\,s^{-1}}$, leading to the largest mixing ratios in the upper atmosphere for this method.

While the differences in the mixing ratios between the run involving a redistribution in a short time period and the runs employing convective area fraction profiles are significant, the two schemes using convective area fraction profiles for the computation of the vertical updraft velocities only show a small difference. Hence, for the $SO_2$-like tracer, the scheme is robust with respect to the particular parameterization of the vertical updraft velocities, as long as the order of magnitude of the
velocities is correct.

We will briefly discuss implications of the differences in the simulations of short-lived species in the model runs and stress their scientific relevance in modelling the time spent in convective updrafts. A more quantitative assessment is outside the scope of this study and is planned for future studies.

Differences in $SO_2$ in the upper troposphere can have an impact on the radiation balance of the Earth and on stratospheric ozone depletion, since they affect the stratospheric aerosol layer (e.g. Rollins et al., 2017). The lower transport of $SO_2$ into the stratosphere in our scheme compared to a scheme with a redistribution in a fixed time step implies a lower contribution of $SO_2$ to the stratospheric aerosol layer, and hence e.g. a lower impact of changes in $SO_2$ emissions in India or China on the stratospheric aerosol layer. A quantitative assessment of this effect, however, is challenging due to large uncertainties in measurements (e.g. Rollins et al., 2017), chemistry and microphysics (e.g. Kremser et al., 2016).

$SO_2$ is a pollutant mainly produced by anthropogenic sources, which is amongst others responsible for atmospheric acidification and the direct and indirect aerosol effect (e.g. Feichter et al., 1996; Berglen et al., 2004; Tsai et al., 2010). Our results suggest that compared to a scheme with a fixed redistribution time step, more $SO_2$ would be converted to $H_2SO_4$ by heterogenous reactions in cloud droplets in the lower troposphere.

Another example for which changes in the convective transport times could be relevant is the contribution of very short-lived bromine substances (VSLS) to the stratospheric bromine budget, which is relevant for stratospheric ozone depletion (e.g. Hossaini et al., 2012). While the lifetime of most VSLS (e.g. $CH_3Br$, $CH_2Br_2$) is too long to be of relevance here, changes of the convective transport times may be relevant for inorganic product gases produced by the VSLS, which are susceptible to washout (e.g. HBr, HOBr) (e.g. Schofield et al., 2011; Hossaini et al., 2012; Wales et al., 2018).

## 6 Conclusions

We present a new Lagrangian convective transport scheme for chemistry and transport models. The scheme is driven by convective mass fluxes and detrainment rates that originate from an external convective parameterization, which can be obtained from meteorological analysis data or general circulation models. The novelty of our method is that we explicitly model the variable time that a trajectory air parcel spends in a convective event by estimating vertical updraft velocity profiles, in contrast to the common approach of a vertical redistribution of air parcels in a fixed time period. Vertical updraft velocities are obtained from combining convective mass fluxes from the meteorological analysis data with a parameterization of convective area fraction profiles. Convective area fractions are obtained by two different parameterizations: a parameterization using a constant convective area profile as well as a parameterization which uses randomly drawn profiles to allow for variability.

We performed simulations with the convective transport model implemented into the trajectory module of the ATLAS chemistry and transport model (e.g. Wohltmann and Rex, 2009), which were driven by ECMWF ERA Interim reanalysis data (Dee et al., 2011).

Our scheme is able to reproduce the convective mass fluxes and detrainment rates from the meteorological analysis data within a few percent. Conservation of the vertical mas distribution in a global 15 year trajectory simulation is also within a few percent, with no apparent trend. Frequency distributions of the modelled vertical velocities agree well with wind profiler measurements conducted at Darwin, Australia, for vertical velocities larger than $0.6\,\mathrm{m\,s^{-1}}$. The agreement was markedly better for the parameterization using a randomly drawn convective area fraction profile than for a constant convective area fraction profile.

Global long-time trajectory simulations of radon-222 were performed and compared to observations. The agreement to the measurements is reasonable, given the large uncertainties in emissions and measurements of radon. Uncertainties in emissions, measurements, chemistry and microphysics of short-lived species generally pose a challenge to the validation of simulations of these species, and there is a clear need to improve on this situation (as also noted by e.g. Forster et al., 2007).

An accurate simulation of the time spent in clouds is important for correctly simulating the chemistry of short-lived species in the troposphere and may be crucial for determining their mixing ratios in the upper troposphere (e.g. Hoyle et al., 2011). As an example for a species for which this is relevant we consider $SO_2$, which is depleted by fast heterogenous reactions in clouds and by a gas-phase reaction with OH. $SO_2$ transported from the troposphere to the stratosphere is one of the most important contributors to the stratospheric aerosol layer in volcanically quiescent periods (e.g. Kremser et al., 2016). In addition, $SO_2$ is a pollutant mainly produced by anthropogenic sources (e.g. Berglen et al., 2004). Allowing for a variable time that an air parcel spends in convection yields a significant effect on the mixing ratios of an $SO_2$-like tracer compared to assuming a redistribution of air parcels in a fixed time step (cf. Figure 18). Remarkably, the mixing ratio distributions were insensitive to the choice of the parameterization of the convective area fraction profile, as long as the order of magnitude of the implied vertical updraft velocities is correct (cf. Figure 18).

Future work and improvements of the method will include the simulation of downdrafts in clouds as well as extensions for applications in the mid-latitudes. For this work, we largely concentrated on the performance in the tropics, the region of the first application cases.

So far, the scheme has been applied for calculations of ammonia transport (Höpfner et al., 2019). A future study will simulate the transport and chemistry of $SO_2$ to examine the contribution of $SO_2$ to the stratospheric aerosol layer.

*Code and data availability.* The source code is available on the AWIForge repository (https://swrepo1.awi.de/). Access to the repository is granted on request under the given correspondence address. This work is based on the revision 1279 of the version control system. Radon climatological profile data over land for the NH mid-latitude region was obtained from Liu et al. (1984). The aircraft radon measurements of the Moffett field and NARE data are available from the Table 1 in Kritz et al. (1998) and Table 3 in Zaucker et al. (1996), respectively. Vertical wind profiler data are available upon request to Alain Protat (alain.protat@bom.gov.au).

*Author contributions.* IW and RL developed and validated the convection model. MR initiated the model development and contributed to the discussion. GG and KP provided the stochastic model for the convective area fraction and contributed to the discussion. WF contributed to the discussion and provided the radon data. AP provided the Darwin wind profiler data and contributed to the analysis of the vertical velocity comparisons. VL provided the CPOL cloud top heights extracted over the Darwin profiler site. CW produced the dual-frequency vertical air velocity retrievals.

*Competing interests.* The authors declare that they have no conflict of interest.

*Acknowledgements.* This research has received funding from the European Community's Seventh Framework Programme (FP7/2007–2013) under grant agreement no. 603557 (StratoClim). We thank ECMWF for providing reanalysis data and Holger Deckelmann for his support in handling and obtaining the ECMWF data. Thanks go to Benjamin Segger for his work on Figures 1 and 2.

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
