# Peer review of "A Lagrangian convective transport scheme including a simulation of the time air parcels spend in updrafts"

_Geoscientific Model Development, 2019_

## Short Comment (SC1) · 13 Mar 2019

Dear authors,

In my role as Executive editor of GMD, I would like to bring to your attention our Editorial version 1.1:

http://www.geosci-model-dev.net/8/3487/2015/gmd-8-3487-2015.html

This highlights some requirements of papers published in GMD, which is also available on the GMD website in the 'Manuscript Types' section:

http://www.geoscientific-model-development.net/submission/manuscript_types.html

[Figure]

In particular, please note that for your paper, the following requirements have not been met in the Discussions paper:

- "The main paper must give the model name and version number (or other unique identifier) in the title."

- "All papers must include a section, at the end of the paper, entitled 'Code availability'. Here, either instructions for obtaining the code, or the reasons why the code is not available should be clearly stated. It is preferred for the code to be uploaded as a supplement or to be made available at a data repository with an associated DOI (digital object identifier) for the exact model version described in the paper. Alternatively, for established models, there may be an existing means of accessing the code through a particular system. In this case, there must exist a means of permanently accessing the precise model version described in the paper. In some cases, authors may prefer to put models on their own website, or to act as a point of contact for obtaining the code. Given the impermanence of websites and email addresses, this is not encouraged, and authors should consider improving the availability with a more permanent arrangement. After the paper is accepted the model archive should be updated to include a link to the GMD paper."

Therefore, add a name and a version number to the title of your manuscript for the lagrangian convection transport scheme. A name and a version number are important to identify your development and the exact state of the development presented in this paper.

Furthermore note, that the developed source code belongs to the publication. Therefore the exact code as used for this publication should be archived in a permanent archive and, if possible, made available to all users.

Yours,

Astrid Kerkweg

---

## Referee Comment (RC1) · Anonymous Referee #1 · 31 Mar 2019

This paper presents a Lagrangian convective transport scheme for chemistry transport models (CTMs). The scheme is implemented in the ATLAS CTM and idealized and realistic test simulations are shown to compare the performance of different variants of the scheme. The paper also includes some validation with wind profiler measurements in the tropics. Whereas I fully agree that the development of Lagrangian convective transport schemes is important and challenging, I found this paper unclear in many places, the novelty was not apparent to me, and the results are not discussed in depth. In its present form, the paper is unsuitable for publication. I recommend very major revisions (see comments below).

[Figure]

Major comments

A) The novelty of this study is not apparent to me. Which elements of this convective transport scheme are "standard" / have been used in other schemes, and which elements are new? First, my impression was that the use of so-called "random convective area fraction profiles" is novel, but then this goes back to Gottwald et al. (2016) . . . The authors should discuss in greater detail how their scheme differs from existing schemes, e.g., the ones mentioned on p. 2 line 5.

B) Related to A) but more general: there must be many studies about convective tracer transport, but very few of them are referenced and discussed. The simulations of Radon-222 and SO2 are not discussed in the framework of the existing literature.

C) The writing is unclear in many places: - p. 1 line 1: what is meant by "ensemble trajectory simulations"? - p. 2 line 3: the same - p. 2 line 16: explain better what is meant by "instantaneous redistribution" - p. 3 lines 23 and 27: unclear to me what exactly is meant by "meteorological data" - p. 4 section 2.2 and p. 6 section 2.4: is this treatment of entrainment and detrainment "standard", i.e., as in other schemes, or is there some novelty here? - p. 7 line 26: this important statement (?) requires much better explanation; it appears rather problematic that fu is not in agreement with the actual number of trajectories in updrafts. - p. 8 line 18: what type of radar measurements? Since this profile (Fig. 3) is important for this study, it would be important to understand better what it is based on. - p. 10 line 3: I don't think that the character of the method is "random", most likely you mean "probabilistic" or "stochastic".

D) My most important concern is related to the fundamental question of how many air parcels / trajectories are required per reanalysis or GCM grid box in order to care about "updrafts". As is discussed in this study, the area covered by updrafts is relatively small even in a region with active deep convection. Figure 3 shows that this area covers less than 1/100 of a 190 x 190 km2 grid box (p. 8 line 19). To me this strongly indicates that many trajectories are needed per grid box in order to "resolve" / capture at least one

or better several of the assumed updrafts in this grid box. However, if there are only relatively few trajectories per grid box, then it does not make sense to explicitly simulate the ascent in updrafts and the question how long a trajectory resides in an updraft becomes obsolete. If my rough estimation is correct, then all simulations performed in this study have far too few trajectories to capture updrafts: e.g., for the idealized experiments in section 4.1 there are 100'000 trajectories for a 60° x 60° domain. Given the ERA-Interim grid boxes of 2° x 2° used in this study, this means that there are roughly 30 x 30 = about 1000 grid boxes, and therefore there are 100 trajectories per grid box. Since (see above) the convective area is <1/100, there is on average not even 1 trajectory that captures an updraft. Things then get much worse for the Radon experiment (section 4.2) where only about 20 trajectories are initialized per 150 km x 150 km grid box (p. 14 line 11). Given such a model setup, I don't understand the general concept of the updraft residence time used in this study. Maybe this issue is addressed on p. 4 line 4 ("The mass of a trajectory ... is typically much larger than the mass transported in a single convective event"), but I could not understand this sentence. Either a much better explanation of the approach or a strongly increased number of trajectories in required to convince me about the feasibility of the convective transport scheme presented in this study.

E) A problem potentially related to D) (at least in my understanding) is the choice of the simulation timestep. In the examples shown, timesteps of 10 or 30 min (why this difference?) have been chosen. I regard these timesteps as way too large to apply the approach outlined in sections 2.2-2.4: since updraft velocities can be up to 20 m s-1, a timestep of 30 min injects a near-surface air parcel deep into the stratosphere. How can this work?

F) Figure 3 is not properly discussed: how is this profile applied in the extratropics? There it does not make much sense that convection can reach an altitude of 15 km . . . so the profile should be scaled with the local tropopause height. And the values for the convective area fraction, is it correct that they only make sense for a given grid size,

i.e., the values must be adjusted if a model is run at higher / lower resolution? This should be discussed.

G) p. 9 line 7: This "deriving of the frequency distribution" is based on 500 hPa vertical velocity from reanalyses or a GCM. However, quantitatively the vertical velocity field is extremely sensitive to the choice of the reanalysis (e.g., NCEP vs. ECMWF) and even more so on the resolution (e.g., ERA-40 vs. ERA-Interim). Therefore – it seems to me – the frequency distribution must be recalculated each time data is used from a different model / reanalysis. Please discuss. The resulting lookup table is mentioned but nothing is shown. Hence, the reader remains unclear how this works and how the result looks like.

H) Where simulation results are described and interpreted (e.g., p. 15 line 17), the paper is very brief. The reader would like to better understand the differences between the experiments.

I) I must say that I don't understand the so-called "random CAF" scheme. First, the description in Section 3.2 is not clear to me. Then, from Figs. 13 and 14 it looks like "random CAF" differs quite a bit from "constant CAF", but when looking at the tracer experiments (Figs. 9-12, 15), then the two schemes yield almost identical results. Why is this the case? And why then should the reader and in general the CTM user community care about the difference between the two schemes? Minor comments

p. 1 line 15: this last sentence appears totally unrelated to the rest of the abstract. Include what the outcome is of this updraft velocity validation.

p. 1 line 18: "correct" –> "accurate" or "appropriate" since we never know the "correct" value.

p. 2 line 28: no need for future tense "First, we present . . . and introduce . . .".

p. 3 line 14: "and" –> "times"

p. 3 line 31: how does the updraft "dominate" the downdraft mass flux? By intensity?

[Figure]

Integrated over the domain, they must be very similar, given mass conservation.

p. 4 and 6: combine Figs. 1 and 2 as two panels in one Figure.

p. 5 line 9: this sentence is awkward, please rephrase.

p. 5 line 13: "m/s" –> m s-1 (and in other places)

p. 6: why is section 2.4 not directly after 2.2?

p. 10 line 7: I would be curious to see pdf of wu for different regions.

p. 11 line 3: "simplified and non-realistic" –> "idealized".

p. 11: Figure 4 is not discussed at all.

p. 13: combine Figs. 6 and 7 as two panels in one Figure.

p. 15: the order of the sections is somehow strange: 4.3 would be better after 4.1 and 4.2 and 4.4 are also somehow related.

p. 16: combine Figs. 9-12 as four panels in one Figure.

p. 20 lines 3 and 13: sentences should not start with "i.e." or "e.g.".

p. 20 line 2: why does the random CAF scheme lead to higher velocities? This is not clear to me.

p. 22: Figure 15 clearly shows the most relevant and interesting result of the paper. I understand that no observations are available to verify these profiles, but I think a more detailed discussion of these profiles is important. The differences are fairly large. What does this imply for tropospheric chemistry? How would the results look like if using a convective transport scheme as implemented in other CTMs or in FLEXPART?

---

## Referee Comment (RC2) · Anonymous Referee #2 · 1 Apr 2019

**General comments**

1. The manuscript introduces a convection scheme for the off-line Lagrangian parcel-based chemistry-transport model ATLAS. The specialty of the scheme is that it resolves temporarily the process of upward transport in a convective updraft, with the aim of better representing chemical transformations, such as the heterogeneous oxidation of sulphur dioxide.

2. The objective and the potential value of such a scheme are obvious, and the step of introducing a corresponding formulation into the ATLAS model is laudable. However, from the work presented it becomes obvious that validation, and specifically the validation of the core component—the residence times during convective updrafts—is very difficult. Therefore, the claim of the paper of a successful validation appears to be not sufficiently supported.

3. The usefulness of the scheme in the context of the whole model will also depend on how well the chemical environment inside a convective cloud is actually modelled. The manuscript is not giving much attention to this aspect, which probably depends strongly on the model resolution (i.e. number of Lagrangian parcels). In addition, it should be compared to the option of just parameterising key reactions such as the heterogeneous oxidation in convective clouds.

4. Admitting that the validation problem is largely inherent and not easily overcome, I think the paper could be acceptable if it would limit itself to a description of the algorithm implemented together with tests conducted so far, while including a clear characterisation of the limitations and the way how a more robust testing and/or tuning will be done, and making it at least plausible that the scheme will be superior to simpler alternatives. This should include, for example, application to case studies with aircraft measurements available.

**Specific comments**

1. It would be good to include a brief introduction to the ATLAS model and how it works, so that the paper can be understood well without first reading other papers, as there is no easy or natural method to include complex chemistry into a Lagrangian model.

2. Page 4 L 1 ff: '*In the following, it is assumed that the mass associated with a trajectory is equal to the mass of the other trajectories and remains constant.*
[Figure]

*This implies that for global model runs, the trajectories need to be distributed uniformly over pressure. The mass associated with the trajectory is then given by air density at the trajectory location and the volume it occupies.*' These sentences are not sufficiently precise, for example, it is not possible to speak about the mass of a trajectory.

3. Figures 1 and 6: The blue colour does print well.

4. Page 5, Eq 4: The equation of state should contain moisture (for example in the form of virtual temperature).

5. Page 5, Eq 5 ff: In $\Delta z_{\mathsf{conv}}$, one would better use just $c$ as subscript, like for other variables.

6. Page 6 Eq. 7 ff: In the integration boundary $z_{\mathsf{start}}$, one would better write $z_0$ or $z_1$. The same holds for variable $M$. In Eq. 9, the subscript 'detrain' could be replaced by $z_2$, $z_d$ or similar. Better not to use (long) words as subscripts.

7. Page 10, L 22: It is not clear why an artifically degraded resolution of 2° is used for the meteorological input from ERA-Interim.

8. Figure 4 and others: It would be good to frame figures (with tick marks on the upper and right axis) and to use secondary ticks as appropriate (in Fig 4, for each day). The number of digits given should not vary along one axis (as it does in Fig. 6 and others).

9. Page 14, L 10-11: I am wondering why trajectories were initialised at random positions rather than on an equal-area grid. Also, the '150 km horizontal resolution' seems to be add odds with a random positioning.

10. Page 14, L 28 ff: '*Radon is distributed evenly over these parcels by assuming a well-mixed boundary layer*' Wording is not good. Eq. 13 is not an equation. The

emission rate would better not be denoted by $e$ in a context where thermodynamic variables appear, it might be confused with vapour pressure. It is also interesting to learn at this place that parcels transport volume mixing ratios, whereas in other places it was said that they represent masses.

11. Page 14-15, para. starting with line 33: The argument is not very clear. It would appear that an artificial minimum boundary-layer height of 500 m would systematically overestimate the input of Rn into the free atmosphere over land during winter, where probably the emission is already overestimated because of the snow cover effects.

12. Page 15 L 17: I would not call this agreement 'reasonable'. Especially in Fig. 11 it is not good. One is also wondering why no comparisons with single flights were done – in the 1990ies there are ERA-Interim data.

13. Page 16, Figure 8: It is not clear what 'Points per layer' means.

14. Page 16 ff, Figures 9-12: It would be more instructive to show mixing ratios rather than concentrations.

15. Page 18 L 9 ff: Do not repeat explanation of the colour of curves in the text.

16. Page 18 ff, Section 4.3: The implications of choosing a specific cut-off value for the vertical velocity need to be discussed. Would it help to use cumulative frequency distrbutions rather than probability densities?

17. Page 21, Figure 14: A step function or just symbols should be used, not continuous curves, as the data represent binned values. The same holds in principle also for Figure 13, but because of its small size, it would probably not make a visual difference.

18. Page 22, L 15-16: The Rn simulation is not suitable to demonstrate the proper long-term stability of mass distribution as radon has a short lifetime.

19. Language

    (a) Authors should pay more attention to upper vs. lower case. One would not normally capitalise 'Chemistry and Transport Model' or chemical elements ('Radon').

    (b) Page 2 L 2: It is surprising to see species in a CTM called 'tracers'.

20. Code and data accessibility: The source code could not be reviewed as it is not anonymously available. Also, to my understanding, access through personal contacts does not conform with GMD code and data availability guidelines. It would also be nice if authors make available the old measurement data on-line in digital form (in which they must have them already), if it is legally possible, rather than pointing to printed publications.

---

## Author Comment (AC1) · 29 Aug 2019

We thank the referee for taking the time to read our manuscript and for their helpful comments!

**General changes**

- We have considerably changed the text throughout the manuscript to improve the logical order of the text and to improve the explanations and comprehensibility. We added new subsections and improved the use of the English language.

**Major comments**

- A) **The novelty of this study is not apparent to me.**

  The novelty is the explicit simulation of the upward transport of air parcels inside convective updrafts and of the variable residence time of air parcels in convection, in contrast to schemes which only redistribute air parcels from the entrainment to the detrainment locations in a fixed time step.

  **Which elements of this convective transport scheme are standard, and which elements are new?**

  The explicit simulation of the upward transport of air parcels inside convective updrafts is new, and the algorithm for detrainment has to be changed accordingly. The redistribution according to entrainment and detrainment probabilities, respectively, is standard. We have more clearly stated this in the description of the algorithm. See also the reply to two comments in major comment C below: comment to page 4, section 2.2 (entrainment) and comment to page 6, section 2.4 (detrainment).

  **First, my impression was that the use of so-called random convective area fraction profiles is novel, but then this goes back to Gottwald et al. (2016)**

  The stochastic parameterisation described in Gottwald et al. had so far not been implemented to estimate convective mass fluxes in convective transport models. The implementation of their method in a transport model is novel.

  **The authors should discuss in greater detail how their scheme differs from existing schemes, e.g., the ones mentioned on p. 2 line 5.**

  The scheme extends the approach in existing schemes by modelling vertical updraft velocities and the time that an air parcel spends inside the convective event. Apart from that, the convective transport part of all schemes (including our scheme), is similar (that is the redistribution of the air parcels given the entrainment rates, detrainment rates and mass fluxes). We hope that we have now more clearly stated the novelty and differences in the introduction and the description of the method.

- B) **There must be many studies about convective tracer transport, but very few of them are referenced and discussed.**

  **The simulations of Radon-222 and SO2 are not discussed in the framework of the existing literature.**

  We have added discussion to section 4.4 (section 4.2 in the original manuscript) on how well the results of other studies compare to radon measurements to put the comparison of our model to radon measurements into perspective. Other studies show differences between their models and the radon measurements of a similar order of magnitude (Jacob et al., 1997, Collins et al., 2002, Forster et al., 2007, Feng et al., 2011). More discussion of the validation of convective transport models was added in the introduction and in the conclusions. The large uncertainties in emissions, measurements, chemistry and microphysics of short-lived species generally pose a challenge for the validation of the simulation of these species, which we think is an important issue. We have added Feichter and Crutzen (1990) as an additional study to the references.

  We added a discussion of the implications of the differences in the simulation of $SO_2$ in the different sensitivity runs to section 5. In addition, we added a paragraph discussing very-short lived bromine species to show that the algorithm is also relevant for species other than $SO_2$.

  This is a technical paper presenting a new algorithm, which is intended as a technical reference to cite when this algorithm is used in an application. It is outside the scope of this paper to give a more detailed discussion of studies of convective tracer transport. Several review papers are cited in the manuscript for reference (e.g. Mahowald et al., 1995, Jacob et al., 1997, Hoyle et al., 2011).

  *Changes to the manuscript:* Added discussion in the introduction and conclusions of the validation by radon and the issue that the uncertainties in measurements, chemistry, microphysics and emissions pose a challenge for the validation of the simulation of short-lived species. Added discussion to section 4.4 (section 4.2 in the original manuscript) on how well other models compare to the radon measurements. Added an additional reference (Feichter and Crutzen, 1990). Added discussion of the implications of the differences in the simulation of $SO_2$ to section 5 and added three new references (Feichter et al., 1996, Kremser et al., 2016, Rollins et al., 2017). Added discussion of very short-lived bromine species to section 5 and added three references (Hossaini etal., 2012, Schofield et al., 2011, Wales et al., 2018).

- C) **Page 1 line 1 and page 2, line 3: What is meant by ensemble trajectory simulations?**

  We agree that it was not obvious what was meant.

  *Changes to the manuscript:* We added the following explanation to the introduction: "In addition, the scheme can be used for applications such as

backward trajectories starting along flight paths or sonde ascents, where it allows for simulating the effect of convection when using a statistical ensemble of trajectories starting at every measurement location."

**Page 2, line 16: Explain better what is meant by instantaneous redistribution**

The Lagrangian convective transport schemes cited here use a short fixed time step to redistribute air parcels, which is not necessarily the same as the advection time step. Collins et al. use a fixed time step of 15 minutes for convection and of 3 hours for large-scale advection. That is, the time period between entrainment and detrainment is fixed to 15 minutes. Forster et al. also use a 15 minute time step. Rossi et al. use a time step of 30 minutes.

*Changes to the manuscript:* We rephrased several parts of the abstract and the introduction to make that more clear. We replaced all occurences of "instantaneous redistribution" by "redistribution in a fixed time step" to avoid misunderstandings.

**Page 3, lines 23 and 27: Unclear to me what exactly is meant by "meteorological data"**

*Changes to the manuscript:* We changed "meteorological data" or "meteorological analysis" to "meteorological analysis data" to make that more consistent throughout the paper and to make clear that we are referring to the same data.

**Page 4, section 2.2: Is this treatment of entrainment [. . .] standard, i.e., as in other schemes, or is there some novelty here?**

Yes, this part of the algorithm is standard, see e.g. Collins et al., 2002 and Forster et al., 2007.

*Changes to the manuscript:* We have added these references to the text.

**Page 6, section 2.4: Is this treatment of [. . .] detrainment standard, i.e., as in other schemes, or is there some novelty here?**

This part of the algorithm is not standard, since it explicitely simulates the upward transport of the air parcel inside the cloud. In other models, only the probability that an entrained air parcel detrains at a given altitude is calculated. The final probability that an air parcel detrains at a certain altitude is the same in our approach and the approach of Collins, Forster and Rossi.

*Changes to the manuscript:* Added to section 2.5 (section 2.4 in the original manuscript): "The approach for detrainment described above differs from the approach employed in previous Lagrangian convective transport schemes, since it takes into account the explicit simulation of the time that air parcels spend in convective updrafts, whereas schemes such as those employed in Collins et al. or Forster et al. assume a constant time that parcels spend in convection. The probability that an entrained air parcel detrains at a given altitude, however, is the same in both approaches."

**Page 7, line 26: This important statement (?) requires much better explanation; it appears rather problematic that fu is not in agreement with the actual number of trajectories in updrafts.**

This was not discussed properly and would leave the reader with the impression that there is a significant problem, which actually is not the case. $f_{up}$ is very small, and the results of the validation runs show that the mass conservation is not noticeably affected by the uncertainty in the number of trajectories in convection.

As an alternative to $f_{up}$, the fraction of trajectory air parcels that are currently in convection in the model run could be used. This is however only possible for global runs. The mass flux of trajectories through a given surface is not necessarily balanced for non-global ensembles of trajectories. The approach would require to average the results over a volume that is small enough to allow for variations in the fraction, but large enough to contain a sufficient number of air parcels.

Another alternative would be to subside all air parcels and not only the air parcels, which are currently not in convection (see Collins et al., 2002). Subsiding air parcels which are currently in convection is however not only unphysical, but also can result in air parcels that descend while they are in convection and that possibly detrain at a lower altitude than they were entrained.

*Changes to the manuscript:* Extended discussion in section 2.6 (section 2.5 in the original manuscript) along the lines outlined above.

**Page 8, line 18: What type of radar measurements? Since this profile (Fig. 3) is important for this study, it would be important to understand better what it is based on.**

*Changes to the manuscript:* We added that the radar is a "precipitation radar" and that the profile is based on the data of two wet seasons (2005/2006 and 2006/2007). We added that the method to obtain the area fractions is "estimating the fraction of convection by comparing the area of convective precipitation to the total measured area".

**Page 10, line 3: I don't think that the character of the method is "random", most likely you mean "probabilistic" or "stochastic"**

*Changes to the manuscript:* Changed.

- D) **My most important concern [. . . ] How many air parcels / trajectories are required per reanalysis or GCM grid box in order to care about updrafts? [. . . ]**

There is a misunderstanding here, namely that the convective updraft area is needed to calculate the number of trajectories affected by convection in a given time period or the probability for a trajectory going into convection, which is not the case! Possibly, this misunderstanding was caused by the formulation "since a grid box contains several convective systems that only

cover a small fraction of the grid box, a statistical approach is necessary"
(page 3, lines 1–2). This was misleading and has been rephrased.

The convective area fraction is not needed for calculating entrainment and
detrainment probabilities and the probability is independent of the area
covered by convection. It is *only* needed for the calculation of the vertical
updraft velocities. Hence, it is not used in the descriptions of the other
Lagrangian convective transport schemes (Collins, Forster, Rossi).

The quantity which is relevant for the entrainment probability is the en-
trainment rate integrated over altitude (with most entrainment typically
at cloud base) and not the convective area fraction (see also discussion in
section 2.2 of the original manuscript and Equation 3). It is only relevant
how much air can be processed by entrainment in a given time period
compared to the mass of the grid box. The probability of convection is
therefore also dependent on the considered time period.

While the mass flux of the entrained air is proportional to the product of
convective area fraction and vertical updraft velocity (see Equation 4 and
discussion), these quantities are not needed for the calculation of the prob-
ablities, which only depend explicitly on the entrainment rate. A small
updraft velocity and a large convective area and a large updraft velocity
and a small convective area lead to the same result for the entrainment
rate.

The only place where convective area fractions are needed in the model
are the vertical updraft velocities, which cannot be deduced from the
mass fluxes alone. The only reason for this is that the mass fluxes in ERA
Interim are given as grid-box means, while the mass flux inside the cloud
is needed.

To show that the number of trajectories is sufficient to capture the up-
drafts, we calculated a frequency distribution of the probability that a
trajectory is entrained into a convective cloud for all trajectories below
2 km from the first time step of the run in the tropical Pacific described in
section 4.1. 77 % of the trajectories have a probability greater than zero
to entrain into a convective cloud in a time period of 10 minutes. The
mean probability for entrainment for an individual trajectory (including
zero values) is 1 % and the maximum value is 13 %. The figure on the next
page shows the frequency distribution.

The trajectories which have a probability greater than zero to entrain are
distributed over about 1000 grid boxes. About 20 trajectories per grid
box have an average chance of more than 1% (each) of entraining into a
convective cloud within 10 minutes. It is clear from these numbers that
not only at any given point in time, there is large number of trajectories
capturing an updraft, but also that all individual grid boxes are covered
well.

[Figure]

*Changes to the manuscript:* Changed formulation at page 3, lines 1–2 to: "Typical resolutions of meteorological analysis data are of the order of 1° x 1°. A grid box of the analysis typically contains several convective systems which only affect a small fraction of the mass contained in the grid box, which necessitates a statistical approach."

**Maybe this issue is addressed on p. 4 line 4 ("The mass of a trajectory [...]")**

Part of the issue is addressed here. The equations of the model are independent of the mass of the trajectory air parcel (for example, Equation 3). Thus, in a global model where the trajectories fill the model domain, a larger mass associated with a trajectory parcel (i.e. a lower density of trajectories per volume) leads to a lower number of trajectories in convection at a given point in time, which balances the higher mass moved per convective event.

Also, in response to a comment of the other reviewer, we considerably rephrased and extended the paragraph.

*Changes to the manuscript:* We considerably extended the discussion at the end of section 2.1 and moved the discussion to a new section 2.2 (in response to the other reviewer).

- E) **In the examples shown, timesteps of 10 or 30 min (why this difference?) have been chosen. I regard these timesteps as way too large to apply the approach outlined in sections 2.2–2.4: since updraft velocities can be up to 20 m $s^{-1}$, a timestep of 30 min injects a near-surface air parcel deep into the stratosphere. How can this work?**

The simulation time step inside the convective event is 10 seconds and not 10 minutes (e.g. original manuscript page 3, lines 12–15 and page 5, lines 9–14). The choice of the timestep is discussed under consideration of the updraft velocities on page 5, lines 13–14.

We are aware that the two time steps for the large scale advection outside convection ($\Delta t$) and for the updraft inside convection ($\Delta t_{conv}$) can easily be confused. We have now clarified some of the notation.

*Changes to the manuscript:* Clarified the notation. In particular, we have changed "trajectory time step" consistently to "advection time step of the trajectory model" and changed "intermediate time step" consistently to "convective intermediate time step".

**... timesteps of 10 or 30 min (why this difference?)...**

The difference is due to computational constraints. The long-time run comprises more than 15 years. Simulation time is considerably reduced by changing the time step from 10 min to 30 min without changing the results significantly (the time step is still much shorter than the lifetime of radon).

1-year runs with a time step of 10 minutes, 0.75° x 0.75° resolution of the analysis and a mean distance of the trajectories of 75 km have been performed to demonstrate that the results do not change significantly. They show that the runs with a time resolution of 30 min, a horizontal resolution of 2° x 2° and a mean distance of 150 km give nearly identical results (see figure, left: 2° x 2°, 30 min from Fig. 10 manuscript, right: 0.75° x 0.75°, 10 min).

[Figure]

In response to a comment of the other reviewer, we increased the resolution of the ERA Interim reanalysis to 0.75° x 0.75° for the high resolution run. In addition, the runs from section 4.1 and the $SO_2$ run are based on ERA Interim 0.75° x 0.75° analysis data now.

*Changes to the manuscript:* Added to section 4.4 (section 4.2 in the original manuscript): "The change from 10 minutes to 30 minutes and from 0.75° x 0.75° to 2° x 2° is due to computational constraints. We performed 1-year test runs with 0.75° x 0.75° resolution, a 10 minute time step and a mean horizontal distance of 75 km of the trajectories that show that the

results of the run with the lower horizontal and time resolution are nearly identical.". Changed the resolution of the ERA Interim data in the runs in section 4.1 and section 5 to 0.75º x 0.75º.

- F) **Figure 3 is not properly discussed: how is this profile applied in the extratropics? There it does not make much sense that convection can reach an altitude of 15 km . . . so the profile should be scaled with the local tropopause height.**

We agree that this was not clear. The scheme was originally developed for an application in the tropics (original manuscript page 8, line 21). Strictly speaking, an application of the algorithm in the extratropics would require a different convective area fraction profile. However, the global long-time simulations of radon are not sensitive to the choice of the convective area fraction profile because of the globally constant lifetime of radon (see explanation in reply to comment I). Hence, using a tropical profile in the radon runs does not noticeably change the results compared to a run using a profile for the mid-latitudes.

*Changes to the manuscript:* We added additional discussion along these lines in section 3.1 and a detailed explanation in new section 4.4.4 (see reply to major comment I).

**And the values for the convective area fraction, is it correct that they only make sense for a given grid size**

This is correct and we agree that it is important to discuss this in section 3.2. The frequency distribution of the measured convective area fractions depends on the domain size of the CPOL radar. The domain size should be comparable to the grid size of the meteorological analysis data to obtain a meaningful distribution of vertical updraft velocities. The full domain size of the radar is $190 \times 190 \, \text{km}^2$, which is comparable to the horizontal resolution of 2º x 2º of the ERA Interim data. As the domain size decreases, the frequency distribution approximates a bimodal distribution: In the limit of domain sizes below typical cloud sizes, the fraction can only be 0 or 1. That is, grid cells completely covered by convection and completely free of convection become more frequent (e.g. Arakawa and Wu, J. Atmos. Sci., 70, 7, 1977-1992, 2013).

It is desirable that the method gives meaningful results for other resolutions than 2º x 2º and can be applied in the range of typical GCM and reanalysis resolutions. In fact, in response to the other reviewer, now also runs with 0.75º x 0.75º resolution are performed.

[Figure]

The figure shows the dependence of the standard deviation of the frequency distribution of measured convective area fractions on the used domain size of the CPOL radar. Results are shown for domain sizes of $190 \times 190\,\mathrm{km}^2$, $100 \times 100\,\mathrm{km}^2$ and $50 \times 50\,\mathrm{km}^2$. For the smaller domain sizes, the measurement domain of the radar has been divided into smaller subdomains. It is evident that the frequency distributions for different domain sizes differ significantly.

The current implementation of the algorithm does not consider this effect, and it is not clear if incorporating a distribution of the convective area fractions which depends on the grid size would lead to a significant change of the results of trajectory runs or not. An implementation of frequency distributions of the convective area fraction that depend on grid size is only planned for a future version, since this would mean a considerable additional effort.

*Changes to the manuscript:* Added discussion to section 3.2 along the lines outlined above. Added figure of the standard deviations for different domain sizes.

- G) **However, quantitatively the vertical velocity field is extremely sensitive to the choice of the reanalysis (e.g., NCEP vs. ECMWF) and even more so on the resolution (e.g., ERA-40 vs. ERA-**

**Interim). Therefore — it seems to me — the frequency distribution must be recalculated each time data is used from a different model / reanalysis. Please discuss.**

This is a good point. It is important for the method that the large-scale vertical velocities from the Darwin/Kwajalein dataset and the large-scale velocities from the reanalysis used for the trajectory calculations have a similar distribution.

The figure shows the frequency distributions of the vertical velocity at 500 hPa from the Darwin dataset, ERA Interim and NCEP, and additionally, two different horizontal resolutions for ERA Interim (0.75º x 0.75º and 2º x 2º resolution). For the reanalysis data, the vertical velocity at 500 hPa at all grid points between 180º E and 240º E and 30º S and 30º N for the arbitrary date 1 June 2010 is used. The frequency distribution of the large scale vertical velocities of the Darwin dataset compares sufficiently well with the frequency distribution of the reanalyses and differences are acceptable in view of other uncertainties of our method, e.g. the uncertainties of the convective area fraction.

[Figure]

Hence, we did not apply a scaling or other correction to the large-scale vertical velocities from ERA Interim. But there may be cases where the vertical velocities from different reanalysis datasets have to be shifted or scaled to obtain a realistic distribution of the convective area fractions.

*Changes to the manuscript:* We added a paragraph and the figure above to section 3.2 and discuss the dependence of the method on the different distributions of the large-scale vertical velocity fields in the different reanalyses.

**The resulting lookup table is mentioned but nothing is shown.**

The figure shows the cumulative distribution of the convective area fraction as a function of the large scale vertical wind, which is used as the lookup table.

[Figure]

*Changes to the manuscript:* Added the figure showing the lookup table to the new manuscript.

- H) **Where simulation results are described and interpreted (e.g., p. 15 line 17), the paper is very brief. The reader would like to better understand the differences between the experiments.**

We expanded the discussion of the radon runs in section 4.4.4 (section 4.2 in original manuscript). We added that the runs with convection generally show higher radon concentrations than the runs without convection in the middle and upper troposphere due to the fast transport of radon from the boundary layer to the detrainment level. A more detailed interpretation of the profiles is however difficult due to the large-scale horizontal averaging.

We added additional discussion to section 4.4 (section 4.2 in the original manuscript) on how well the results of other studies compare to radon measurements to put the comparison of our model to radon measurements into perspective. Other studies show differences between their models and the radon measurements of a similar order of magnitude (see major comment B).

A discussion of the implications of the results of the $SO_2$ runs and of the scientific relevance of developing a convection model which simulates the time spent in updrafts was added: We added a discussion of the implications of the differences in the simulation of $SO_2$ in the different sensitivity

runs to section 5 and a paragraph discussing very-short lived bromine species to show that this is also relevant for other species than $SO_2$.

*Changes to the manuscript:* Expanded the discussion of the radon runs in section 4.4.4. Added discussion to section 4.4 (section 4.2 in the original manuscript) on how well the results of other studies compare to radon measurements. Added discussion of the implications of the results of the $SO_2$ runs to section 5. Added a paragraph discussing very-short lived bromine species to section 5.

- I) **I must say that I don't understand the so-called "random CAF scheme". First, the description in Section 3.2 is not clear to me.**

Vertical updraft velocities are obtained from combining convective mass fluxes from meteorological analysis data with a parameterization of convective area fraction profiles. We implement two different parametrizations for the convective area fraction, a parametrization using an observed constant convective area fraction profile as well as a parametrization which uses randomly drawn profiles to allow for variability in the convective area fractions. We rephrased the abstract, introduction and conclusions to make that more clear and rephrased section 3.2 to provide a more detailed explanation.

Furthermore, we hope that the reply to comment F (dependence of convective area fraction on grid size) and comment G (dependence of large scale vertical velocity on reanalysis, figure showing lookup table) and the additional discussion in section 3.2 make it more clear what has been done.

*Changes to the manuscript:* Rephrased abstract, introduction, section 3.2 and conclusions along the lines outlined above.

**Then, from Figs. 13 and 14 it looks like "random CAF" differs quite a bit from "constant CAF", but when looking at the tracer experiments (Figs. 9–12, 15), then the two schemes yield almost identical results. Why is this the case?**

The reason for the almost identical results for the radon simulations is that the lifetime of radon is globally constant. For a tracer with a globally constant lifetime, it makes no differences if it was transported slowly upwards from the emission at the boundary layer to 10 km in the last 10 days or if it first was transported quickly by convection to 10 km within one hour, and then stayed at 10 km for 9 days and 23 hours. The amount of radon that decays only depends on the time passed since the last contact with the boundary layer, when it was emitted (see original manuscript page 15, lines 21–26 and new section 4.4.4).

Differences have to be expected for the $SO_2$-like tracer. These differences are relatively small in our model runs, which means that the results are insensitive to the uncertainties in the parameterization of the vertical updraft velocities.

*Changes to the manuscript:* We added additional discussion of the effect of globally constant lifetimes along the lines outlined above to section 4.4.4.

**And why then should the reader and in general the CTM user community care about the difference between the two schemes?**

It is not implied in the text that the community should care about the difference. It is a valid approach to try out several approaches in a new algorithm and to see what works best or if several approaches yield similar results.

**Minor comments**

- **page 1, line 15: this last sentence appears totally unrelated to the rest of the abstract. Include what the outcome is of this updraft velocity validation.**

  The sentence was directly related to the preceding sentence, which mentioned the validation of the mass conservation and validation with radon.

  *Changes to the manuscript:* We rephrased the abstract to include the main results of the validation.

- **page 1, line 18: "correct" $->$ "accurate" or "appropriate"**

  *Changes to the manuscript:* Changed.

- **page 2, line 28: no need for future tense**

  *Changes to the manuscript:* Changed.

- **page 3, line 14: "and" $->$ "times"**

  *Changes to the manuscript:* Changed to "multiplied by".

- **page 3, line 31: How does the updraft dominate the downdraft mass flux? By intensity? Integrated over the domain, they must be very similar, given mass conservation.**

  This is a misunderstanding caused by the confusion of the downdraft mass flux in the cloud with the slow subsidence outside of the cloud. The subsidence outside the clouds has to balance the convective mass flux inside the clouds (sum of updrafts and downdrafts), see section 2.6 (2.5 in the original manuscript).

  *Changes to the manuscript:* We added the phrases "updraft inside clouds", "downdraft inside clouds" and "subsidence outside clouds" at some additional locations.

- **page 4 and 6: combine Figs. 1 and 2 as two panels in one Figure**

  We would like to keep the separate figures. We do not see a benefit in combining the figures.

- **page 5, line 9: this sentence is awkward, please rephrase.**

  *Changes to the manuscript:* Split up into two sentences: "If a parcel is marked as taking part in convection, it is transported upwards for the vertical distance that it will be able to ascend in one intermediate convective time step $\Delta t_{\mathrm{conv}}$ (10 seconds). The vertical distance is determined by the vertical convective updraft velocity."

- **page 5, line 13: "m/s" $- > \mathrm{m\,s^{-1}}$**

  *Changes to the manuscript:* Changed throughout the manuscript.

- **page 6: why is section 2.4 not directly after 2.2?**

  This is the natural temporal order of the events: 2.2 entrainment, 2.3 upward transport, 2.4 detrainment. This is also the order of the steps in the algorithm (see original manuscript page 3, lines 11–16).

- **page 10, line 7: I would be curious to see pdf of wu for different regions.**

[Figure]

  The plot shows the pdf of the vertical updraft velocities derived from ERA Interim (model level 21, corresponding to about 520 hPa, June 2010) for four different regions: Pacific (180–240º E, 15º S–15º N), Atlantic (330–345º E, 15º S–15º N), Africa (0–45º E, 15º S–15º N), South America (285–315º E, 15º S–15º N). There are no significant differences for velocities below about 7 m/s. The percentage of velocities > 20 m/s is lower than 0.1 % for all regions.

- **page 11, line 3: "simplified and non-realistic" $- >$ "idealized"**

  *Changes to the manuscript:* Changed.

- **page 11: Figure 4 is not discussed at all.**

  This is only intended as an example, and we feel that a short description is sufficient.

- **page 13: combine Figs. 6 and 7 as two panels in one Figure.**

  See comment to page 4 and 6 above.

- **page 15: the order of the sections is somehow strange: 4.3 would be better after 4.1 and 4.2 and 4.4 are also somehow related.**

  *Changes to the manuscript:* Changed as requested. Moved section 4.2 (original manuscript) to the end of section 4. Section 4.2 (original manuscript) is now section 4.4 (new manuscript), section 4.3 (original manuscript) is section 4.2 (new manuscript) and section 4.4 (original manuscript) is section 4.3 (new manuscript). Divided 4.4 into additional subsections.

- **page 16: combine Figs. 9-12 as four panels in one Figure.**

  See comment to page 4 and 6 above.

- **page 20, line 3 and 13: sentences should not start with "i.e." or "e.g."**

  *Changes to the manuscript:* Changed to "That is" and "For example", respectively.

- **page 20, line 2: why does the random CAF scheme lead to higher velocities? This is not clear to me.**

  The fact that the vertical updraft velocities are typically larger when a randomly drawn convective area fraction profile is used can be readily understood qualitatively: Assuming that $M$, $T$ and $p$ are fixed, the mean updraft velocity in case of a mean constant convective area fraction profile $\langle f_{\mathrm{up}} \rangle$ is simply $\langle w_{\mathrm{up1}} \rangle = \frac{MRT}{\langle f_{\mathrm{up}} \rangle p}$, where $\langle \ldots \rangle$ denotes the mean over all air parcels. In the case of a varying randomly drawn convective area fraction profile, the mean vertical updraft velocities need to be expressed as $\langle w_{\mathrm{up2}} \rangle = \langle \frac{MRT}{f_{\mathrm{up}} p} \rangle = \frac{MRT}{p} \langle \frac{1}{f_{\mathrm{up}}} \rangle$. Since $\langle \frac{1}{f_{\mathrm{up}}} \rangle \geq \frac{1}{\langle f_{\mathrm{up}} \rangle}$ due to the fact that the harmonic mean is always smaller than the geometric mean, we obtain the relation $\langle w_{\mathrm{up2}} \rangle \geq \langle w_{\mathrm{up1}} \rangle$. This implies that also individual realizations of $w_{\mathrm{up}}$ are on average larger for the random convective area fraction profiles.

  *Changes to the manuscript:* Added discussion to section 4.2 (section 4.3 in the original manuscript) along the lines discussed above.

- **page 22: Figure 15 clearly shows the most relevant and interesting result of the paper. I understand that no observations are available to verify these profiles, but I think a more detailed discussion of these profiles is important. The differences are fairly large. What does this imply for tropospheric chemistry?**

We agree that a discussion of the implications of the results of the $SO_2$ runs and of the scientific relevance of developing a convection model which simulates the time spent in updrafts is important. We added a discussion of the implications of the differences in the simulation of $SO_2$ in the different sensitivity runs to section 5. In addition, we added a paragraph discussing very-short lived bromine species to show that this is also relevant for species other than $SO_2$.

*Changes to the manuscript:* We extended the discussion in section 5 by adding paragraphs discussing the implications of the changes in the $SO_2$ simulations and a paragraph discussing very-short lived bromine species as an example for another species for which this could be relevant.

**How would the results look like if using a convective transport scheme as implemented in other CTMs...**

This is a question we are also interested in. We added discussion of how well the results of other models compare to radon measurements in section 4.4.4. A detailed comparison study of several convective transport models is outside the scope of this technical presentation of an algorithm. This would mean a considerable additional effort.

Differences between different models in other studies will often mainly be due to differences in the underlying convective parameterization (see e.g. Feng et al., 2011). This is however a very extensive and difficult topic (e.g. Arakawa, 2004), which is outside the scope of this study.

*Changes to the manuscript:* We added some discussion of how well the results of other models compare to radon measurements in section 4.4.4.

**...or in FLEXPART?**

FLEXPART does not provide single trajectories as output which one could use to run a box model. We are restricted to the build-in simplified chemistry schemes, which are an exponential decay with a fixed lifetime and a simple OH scheme (e.g. Pisso et al., Geosci. Model Dev., doi:10.5194/gmd-2018-333). Hence, it is not possible to do a meaningful comparison due to constraints in FLEXPART.

---

## Author Comment (AC2) · 29 Aug 2019

We thank the referee for taking the time for reading our manuscript and their helpful comments!

**General changes**

- We have considerably changed the text throughout the manuscript to improve the logical order of the text and to improve the explanations and comprehensibility. We added new subsections and improved the use of the English language.

**General comments**

- **Combined reply to the following:**

  **2. From the work presented it becomes obvious that validation, and specifically the validation of the core component — the residence times during convective updrafts — is very difficult.**

  **4. Admitting that the validation problem is largely inherent and not easily overcome, I think the paper could be acceptable if it would limit itself to a description of the algorithm implemented together with tests conducted so far, while including a clear characterisation of the limitations and the way how a more robust testing and/or tuning will be done, and making it at least plausible that the scheme will be superior to simpler alternatives.**

  We agree that more discussion of these issues was needed. Currently, the large uncertainties in emissions, chemistry, microphysics and measurements of many short-lived species do not allow for a quantitative assessment whether our scheme improves the simulation of these short-lived species, even if this is suggested by the more realistic simulation of the time spent in convective clouds. Rather, our scheme allows for estimating the uncertainties in the simulation of these species associated with different parameterizations of vertical transport in convective updrafts. These uncertainties generally pose a challenge for the validation of the simulation of short-lived species, and there is a clear need to improve on this situation (as also noted by e.g. Forster et al., 2007).

  In addition, the globally constant lifetime of radon does not allow to validate the parameterization of the time spent in convective updrafts. Nevertheless, currently radon is probably still the species most suitable for the validation of convective transport models, since there is a lack of good alternatives.

  We have added discussion to section 4.4 on how well the results of other studies compare to radon measurements to put the comparison of our model to radon measurements into perspective. Other studies show differences between their models and the radon measurements of a similar order of magnitude (Jacob et al., 1997, Collins et al., 2002, Forster et al., 2007, Feng et al., 2011).

For many physical parameterizations in GCMs and CTMs there is no sufficient data for validation. The only way to make it more plausible that they are superior is to state that the physical assumptions are closer to reality.

*Changes to the manuscript:* We extended the discussion in the introduction and conclusions to discuss the large uncertainties in the validation of short-lived species as outlined above and to discuss the validation with radon. In addition, we added discussion in section 4.4.4 (section 4.2 in original manuscript) how well other models compare to the radon measurements and on the uncertainties in radon emissions, simulations and measurements.

- **2. The claim of the paper of a successful validation appears to be not sufficiently supported.**

We are aware that validation of the model is difficult and paid attention to a careful formulation of the results. The only occasion in the original manuscript, where we speak of an "successful validation" is at page 22, line 14 in the conclusions. This only refers to the technical part of the validation, i.e., mass conservation and reproduction of the convective mass fluxes and detrainment rates from the reanalysis. Since this part of the sentence is not really needed, we deleted it to avoid confusion.

*Changes to the manuscript:* Deleted "The algorithm is successfully validated by showing that" from the sentence.

**3. The usefulness of the scheme in the context of the whole model will also depend on how well the chemical environment inside a convective cloud is actually modelled. The manuscript is not giving much attention to this aspect, which probably depends strongly on the model resolution (i.e. number of Lagrangian parcels). In addition, it should be compared to the option of just parameterising key reactions such as the heterogeneous oxidation in convective clouds.**

The chemistry scheme is a part of the model which is independent from the transport scheme, and we think that a discussion of chemistry schemes is better suited to a separate study, which may for example study the effects that the different model components have in a complete GCM or CTM.

This is a technical paper presenting a new algorithm for a convective transport scheme. While it is certainly very interesting and important, it is out of the scope of this study to perform a detailed comparison of complex chemistry schemes or to discuss the chemistry of short-lived species like $SO_2$ in detail.

This model was originally developed as part of a larger study of the chemistry and transport of $SO_2$ from the troposphere to the stratosphere. An important part of this study is how the numerous uncertainties in $SO_2$

chemistry, convection, transport and microphysics translate into uncertainties in the $SO_2$ mixing ratios. It was decided to split the publication of this study into two papers. The combined study would have been too extensive and it is not a good idea to start a study about $SO_2$ with a long technical description of a convection model.

Unfortunately, a meaningful validation of the model is difficult with these $SO_2$ simulations and measurements. There are so many uncertainties that the results always can be tuned to agree with the measurements.

**Specific comments**

- **1. It would be good to include a brief introduction to the ATLAS model and how it works, so that the paper can be understood well without first reading other papers, as there is no easy or natural method to include complex chemistry into a Lagrangian model.**

  The ATLAS model is a model consisting of several independent modules. In this study, only the trajectory module is used. The chemistry module and the mixing module are not used.

  Radon and $SO_2$-like tracer mixing ratios are calculated with a simple exponential decay and fixed lifetimes. The more sophisticated chemistry model, which is implemented in the full ATLAS model and uses a system of coupled differential equations, is not employed.

  *Changes to the manuscript:* We changed the text in several locations (abstract, introduction, section 4, conclusions) to make clear that only the trajectory module is used. Added that the trajectory module uses a 4th order Runge-Kutta scheme.

- **2. Page 4 L 1ff: These sentences are not sufficiently precise, for example, it is not possible to speak about the mass of a trajectory.**

  It probably was not clear what the discussion was aiming at.

  We agree that there is no natural way to assign a mass to a single trajectory air parcel. One could argue that a trajectory air parcel only refers to an infinitesimal volume and that only intensive quantities like density are well defined for an air parcel, while extensive quantities like mass are not well defined.

  However, in a global model, where the model domain is filled with trajectory air parcels, this looks different. Here, the volume of the model domain can be divided into smaller subvolumes that make up the complete volume. Each subvolume can be associated with a trajectory air parcel, with the air parcel mass given by the product of density of air and air parcel volume. The same constant mass can be assigned to each trajectory air parcel, which implies that the associated volume is increasing with decreasing air density. Since the subvolumes should not overlap to

avoid that the same air volume is counted twice, this means that trajectory air parcels are distributed uniformly over pressure (but exponentially decreasing over altitude).

This is not merely a theoretical consideration, but becomes important when e.g. the global mass of a chemical species is calculated, or the mass flux of a chemical species through a control surface (as the tropopause).

*Changes to the manuscript:* We considerably extended the discussion in section 2.1 as outlined above and moved the discussion to a new subsection 2.2.

- **3. Figures 1 and 6: The blue colour does print well.**

  You probably mean "does not print well"? A darker blue is used now.

- **4. Page 5, Eq 4: The equation of state should contain moisture**

  For a worst case scenario with a temperature of $300\,\mathrm{K}$ and a relative humidity of $100\,\%$, the change in density compared to the dry density is $2.2\,\%$. This is negligible given the uncertainties of the method.

  *Changes to the manuscript:* We added a note to the text.

- **5. Page 5, Eq 5 ff. One would better use just $c$ as subscript.**

  Thanks for noting this. That was inconsistent throughout the manuscript, sometimes $c$ was used, and sometimes "conv".

  *Changes to the manuscript:* We changed the subscript to "conv" consistently (see also below).

- **6. Page 6 Eq. 7 ff: Better not to use (long) words as subscripts.**

  In our opinion, short words as subscripts help to understand the equations. We agree that very long words (e.g. "subsidence") make the equations hard to read.

  *Changes to the manuscript:* We changed all subscripts of all variables consistently to consist of short words.

- **7. Page 10, L 22: It is not clear why an artificially degraded resolution of 2 degrees is used for the meteorological input from ERA-Interim.**

  The difference is due to computational constraints. The long-time run comprises more than 15 years. Simulation time is considerably reduced by changing the resolution from the original resolution of 0.75º x 0.75º to a resolution of 2º x 2º without changing the results significantly.

  The results of the long-time runs are not particularly sensitive to the resolution of the reanalysis data. 1-year runs with a time step of 10 minutes, 0.75º x 0.75º resolution of the analysis and a mean distance of the trajectories of 75 km have been performed to demonstrate that the results do not change significantly (a related comment of reviewer 1 asked for the

difference that the change in time step from 10 min in the simplified run to 30 min in the radon run would cause). The runs with a time resolution of 30 min, a horizontal resolution of 2º x 2º and a mean distance of 150 km give nearly identical results (see figure, left: 2º x 2º, 30 min from Fig. 10 manuscript, right: 0.75º x 0.75º, 10 min).

[Figure]

The idealized runs from section 4.1 and the $SO_2$ run, which comprises a shorter time period, are based on ERA Interim data with a resolution of 0.75º x 0.75º now.

*Changes to the manuscript:* We added discussion of the 1-year runs to section 4.4.1 (section 4.2 in original manuscript). We increased the resolution to 0.75º x 0.75º in the simplified runs in section 4.1 and for the $SO_2$ runs in section 5.

- **8. Figure 4 and others: It would be good to frame figures (with tick marks on the upper and right axis)...**

*Changes to the manuscript:* Done.

**...and to use secondary ticks as appropriate (in Fig. 4, for each day).**

We are sorry that this is not feasible. Our software does only allow automatic placement of secondary tick marks, but there is no control over the spacing.

**The number of digits given should not vary along one axis.**

It is common practice that digits vary. For example, we do not think it makes sense to label the pressures "0800", "0900", "1000" or the mass flux "0.025", "0.030".

- **9. Page 14, L 10–11: I am wondering why trajectories were initialised at random positions rather than on an equal-area grid.**

The random positioning is the default for trajectory initialization in the ATLAS model. It is normally used to avoid that an initialization on a regular grid can have any systematic effect on the results. It was used here for simplicity. An equal-area grid would probably work equally well for the application in this study.

*Changes to the manuscript:* We added that this is the default initialization scheme of ATLAS and that it is normally used to avoid any systematic effects to the paragraph in section 4.4 (4.2 in original manuscript).

**Also, the 150 km horizontal resolution seems to be add odds with a random positioning.**

This indeed needs a better explanation.

*Changes to the manuscript:* We changed the text to "Trajectories are initialized at random positions (both horizontally and in pressure) between $1100\,\text{hPa}$ and $50\,\text{hPa}$. The number of trajectories is chosen in such a way that the mean horizontal distance of the trajectories is $150\,\text{km}$ in reference to a layer of a width of $50\,\text{hPa}$."

- **10. Page 14, L 28 ff: "Radon is distributed evenly over these parcels by assuming a well-mixed boundary layer" Wording is not good.**

*Changes to the manuscript:* Rephrased the sentence to "Radon is emitted into all trajectory air parcels that are in the boundary layer by assuming a well-mixed boundary layer, and a volume mixing ratio $x$ of..."

**Eq. 13 is not an equation.**

*Changes to the manuscript:* Changed the text to "volume mixing ratio $x$" and the equation to $x = \dots$

**The emission rate would better not be denoted by $e$ in a context where thermodynamic variables appear, it might be confused with vapour pressure.**

The disadvantage of using a letter different from $e$ is that the association with the starting letter of "emission" is lost, so this is a compromise. $\varepsilon$ is already used for the entrainment probability in the text, and $E$ is used for the entrainment rate.

**It is also interesting to learn at this place that parcels transport volume mixing ratios, whereas in other places it was said that they represent masses.**

This is no contradiction. The basic assumption behind the concept of an "air parcel" is that it contains the same set of atoms at any given time. It follows that the mixing ratio of a given species is conserved along a trajectory (given that no chemical reactions take place) and that the mass of air is conserved.

- **11. Page 14–15, para. starting with line 33: The argument is not very clear. It would appear that an artificial minimum boundary-layer height of 500 m would systematically overestimate the input of Rn into the free atmosphere over land during winter, where probably the emission is already overestimated because of the snow cover effects.**

Our approach may cause some Radon which would be "trapped" in the boundary layer to end up in the free troposphere in the simulation and may cause some differences of the simulation to the Radon measurements.

However, assuming a minimum boundary layer height (or some similar measure) is unavoidable in global trajectory models, since the required number of trajectories needed for a model run which resolves the boundary layer by far exceeds any reasonable number that is computationally feasible.

The mass of radon emitted into the boundary layer per time period and area is still the same as with the actual boundary layer height and is not overestimated. This is accomplished by dividing by the boundary layer height $z_{\mathrm{BL}}$ in Equation 13.

*Changes to the manuscript:* We added discussion to the paragraph along the lines outlined above.

- **12. Page 15 L 17: I would not call this agreement "reasonable". Especially in Fig. 11 it is not good.**

We agree that a better explanation is needed why the agreement is called "reasonable, given the large uncertainties in measurements and emissions". We think that there are good reasons to keep this formulation.

We have now added discussion to section 4.4 (section 4.2 in the original manuscript) on how well the results of other studies compare to radon measurements to put the comparison of our model to radon measurements into perspective. Other studies show differences between their models and the radon measurements of a similar order of magnitude (e.g. Mahowald et al., 1995, Jacob et al., 1997, Collins et al., 2002, Forster et al., 2007, Feng et al., 2011). This suggests that a better agreement cannot be expected, given the uncertainties in measurement, emission and the simulation. The wording in other studies describing the agreement is comparable. E.g. Feng et al. states that their results "agree reasonably well" to the radon measurements. Their Figs. 13 and 14 show that the differences are comparable. Currently radon is probably still the species most suitable for the validation of convective transport models, since there is a lack of good alternatives.

The underestimation of radon by the simulation in Fig. 11 has also been observed in other studies (e.g. Jacob et al., 1997, Forster et al., 2007). This may be due to uncertainties in emission and due to the fact that measurements from coastal areas are included, where horizontal radon gradients are high and difficult to model (see Forster et al., 2007).

*Changes to the manuscript:* We extended the discussion as outlined above. Discussion was added to the introduction and conclusions, discussion in section 4.4 (4.2 in the original manuscript) was extended, and a discussion of the differences seen in Fig. 11 was added.

**One is also wondering why no comparisons with single flights were done in the 1990ies there are ERA-Interim data.**

The uncertainties of both the simulation and the radon measurements are so large that the data need to be averaged to obtain meaningful results. This is the common approach in most studies (e.g. Forster et al., 2007, Feng et al., 2011).

- **13. Page 16, Figure 8: It is not clear what "Points per layer" means.**

  *Changes to the manuscript:* Changed to "trajectory air parcels per layer".

- **14. Page 16 ff, Figures 9-12: It would be more instructive to show mixing ratios rather than concentrations.**

  The plots show the frequency of radioactive decay events (mBq) per volume ($m^3$), which is proportional to concentrations. This is the standard unit for radon, which is found in the majority of the publications (see e.g. Mahowald et al., 1995, Collins et al., 2002, Feng et al., 2011). For the reason of being comparable to other studies, we would like to stick to the units.

- **15. Page 18 L 9 ff: Do not repeat explanation of the colour of curves in the text.**

  We do not see a disadvantage. We would like to keep the text as is.

- **16. Page 18 ff, Section 4.3: The implications of choosing a specific cut-off value for the vertical velocity need to be discussed.**

  We substantially extended and rephrased this discussion. Part of the problem is caused by the conceptual problem of defining what a convective updraft is in the measurements. It is common to apply a lower threshold to the vertical updraft velocities to define convective situations in the measurements. Typically, this threshold is between $0\,\mathrm{m\,s^{-1}}$ and $1.5\,\mathrm{m\,s^{-1}}$ and may have a significant effect on the results (e.g. Kumar et al., 2015). Note that the $0.6\,\mathrm{m/s}$ cut-off is applied in Fig. 15 only for comparison. It does not appear in the model formulation.

  Replacing the simulated vertical updraft velocities by the measured vertical updraft velocities in the model would increase the average residence time between entrainment and detrainment. In turn, this would lead to a lower concentration of a short-lived species like $SO_2$ in the upper troposphere.

  *Changes to the manuscript:* Substantially expanded and rephrased the discussion in section 4.2 (4.3 in the original manuscript) as outlined above.

  **Would it help to use cumulative frequency distributions rather than probability densities?**

No. Since the cumulative frequency distribution is the integral of the probability density, changes at the small values of velocity will affect the values of the cumulative frequency distribution at large velocities.

- **17. Page 21, Figure 14: A step function or just symbols should be used, not continuous curves, as the data represent binned values.**

In this case the binned data is used to approximate a curve which should be continuous in theory (by using an infinite number of bins). For this plot, which shows 30 bins, there is hardly any difference to a "continuous" curve.

- **18. Page 22, L 15–16: The Rn simulation is not suitable to demonstrate the proper long-term stability of mass distribution as radon has a short lifetime.**

This is a misunderstanding. Radon is not used to demonstrate the long-term stability of the mass distribution. The long-term run is used for two separate purposes: a) To demonstrate the stability of the mass distribution, and b) to validate the model with radon. The radon mixing ratios are not needed to demonstrate the stability of the mass distribution, and the positions of the trajectories are sufficient for this. The stability of the mass distribution is demonstrated by counting the trajectory air parcels in a given altitude layer. Since every trajectory parcel is associated with a constant mass, this is equivalent to determining the mass in a layer.

*Changes to the manuscript:* Changed the text in several locations to avoid misunderstandings: Added a new section 4.4.3 with the title "Conservation of vertical mass distribution". We changed the text in section 4.4.3 (originally section 4.2, page 14, lines 19–24) by including: "We revisit the issue of the conservation of the vertical mass distribution in this more realistic setup (compared to the idealized setup in Section 4.1)". We changed "mass distribution" in the sentence "The number of trajectories ... at the start ... compares well with the mass distribution at the end" to "number of trajectories". Added "conservation of vertical mass distribution of air (not of radon)" to the description in the text.

- **19. a) Authors should pay more attention to upper vs. lower case.**

*Changes to the manuscript:* Changed.

**b) Page 2 L 2: It is surprising to see species in a CTM called "tracers"**

*Changes to the manuscript:* Changed "tracers" to "species".

- **20. Code and data accessibility**

We would be happy to provide the source code to you by creating an account on our repository for you, if you feel this is necessary.

As far as we understand it from the "model and data policy" statement, we are obliged to make the source code available to the editor, so that would have been the designated point of contact to our understanding.

**It would also be nice if authors make available the old measurement data on-line in digital form (in which they must have them already), if it is legally possible, rather than pointing to printed publications.**

We have no permission to do that.

---

## Author Comment (AC3) · 29 Aug 2019

Dear Astrid Kerkweg,

- **The main paper must give the model name and version number (or other unique identifier) in the title**

  This was discussed with the editor Patrick Jöckel before submission. He agreed to the current title. The manuscript presents an algorithm, which in principle can be implemented into any given chemistry and transport model or trajectory model, which we would like to stress by not mentioning a model name in the title.

The implementation into the ATLAS model is done to demonstrate the feasibility of the approach and to perform validation runs, but any other model could have been used for this. ATLAS is a model consisting of several independent modules (trajectories, chemistry, mixing,...), and only the trajectory module is used in this study, which only comprises a small part of the source code. Hence, it would be potentially misleading to prominently state the name of the ATLAS model in the title.

- **Code availablity**

  Please understand that I do not have the permission or authority to change the current statement. But I will happily provide the source code on any reasonable request.

  I agree that is important to know the exact state of development. I have added to the code availability section that the manuscript is based on the revision 1279 of the version control system. The repository allows to retrieve this version without problems.

Best regards, Ingo Wohltmann